# Unveiling Concept Attribution in Diffusion Models

**Quang H. Nguyen**[1], **Hoang Phan**[1,2], **Khoa D. Doan**[1,2]
[1]College of Engineering and Computer Science, VinUniversity
[2]VinUni-Illinois Smart Health Center, VinUniversity
{quang.nh, 21hoang.p, khoa.dd}@vinuni.edu.vn

## Abstract

Diffusion models have shown remarkable abilities in generating realistic and high-quality images from text prompts. However, a trained model remains largely black-box; little do we know about the roles of its components in exhibiting a concept such as objects or styles. Recent works employ causal tracing to localize knowledge-storing layers in generative models without showing how other layers contribute to the target concept. In this work, we approach diffusion models' interpretability problem from a more general perspective and pose a question: *"How do model components work jointly to demonstrate knowledge?"*. To answer this question, we decompose diffusion models using component attribution, systematically unveiling the importance of each component (specifically the model parameter) in generating a concept. The proposed framework, called **C**omponent **A**ttribution for **D**iffusion Model (CAD), discovers the localization of concept-inducing (positive) components, while interestingly uncovers another type of components that contribute negatively to generating a concept, which is missing in the previous knowledge localization work. Based on this holistic understanding of diffusion models, we present and empirically evaluate one utility of component attribution in controlling the generation process. Specifically, we introduce two fast, inference-time model editing algorithms, CAD-Erase and CAD-Amplify; in particular, CAD-Erase enables erasure and CAD-Amplify allows amplification of a generated concept by ablating the positive and negative components, respectively, while retaining knowledge of other concepts. Extensive experimental results validate the significance of both positive and negative components pinpointed by our framework, demonstrating the potential of providing a complete view of interpreting generative models. Our code is available here.

## 1 Introduction

Recent developments in diffusion models [18, 26, 36, 37] have greatly improved the synthesizing capabilities, in terms of image quality and the diversity of generated knowledge. However, these models lack interpretability; we do not fully understand how they can translate simple prompts to coherent visual outputs. To investigate how generative models express learned concepts, a recent line of work studies which components in the model store knowledge [2, 27]. In language models, Meng et al. [27] propose causal tracing to locate layers storing facts and reveal that knowledge is localized in middle-layer MLP modules. Basu et al. [2] transfer this approach to diffusion models and propose the *knowledge distributed hypothesis*. They show that, unlike language models, knowledge is distributed among a set of UNet components and the first self-attention layer of the text encoder. These works shed light on interpreting generative models, enabling more effective model editing [3, 2]. Nevertheless, they focus only on knowledge storage—modules that are responsible for generating concepts—and on coarse-grained components, such as layers. This perspective may overlook more subtle properties and other types of modules that also influence the outputs.

To address that limitation, this paper first poses a more general question: *How do components in diffusion models contribute to a generated concept?* We introduce a framework that *predicts* the model's behavior given the presence of each component by identifying its contribution through an efficient linear counterfactual estimator [35]. Through this framework, called **C**omponent **A**ttribution for **D**iffusion Model (CAD), we advance the understanding of how model components activate concepts (e.g., objects, styles, or explicit contents) in diffusion models. In contrast to the prior work that focuses on the model's layers, CAD allows analysis of more fine-grained components. Specifically, we revisit the *distributed hypothesis* [2] and leverage CAD to identify concept-inducing (or positive) components that are similar to knowledge storage. However, by focusing on the most fine-grained components, i.e., the model's parameters, we propose the *localization hypothesis*: knowledge is localized in a small number of parameters. Surprisingly, besides the positive components, CAD also reveals the existence of components that contribute negatively to generating the target concept, which is missing in the previous studies. Ablating positive or negative components decreases or increases the probability of generating the corresponding knowledge. As one example of its utility, this holistic understanding of diffusion models enables a lightweight model editing capability, i.e., to remove (positive) or recall (negative) a concept. Figure 1 illustrates the proposed CAD framework. In summary, our contributions are:

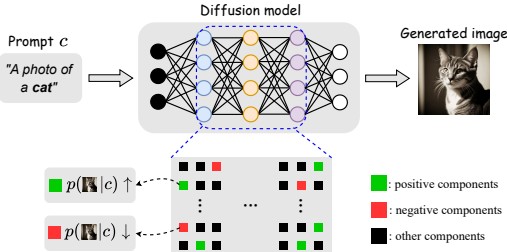

Figure 1: Overview of our framework. We show that there exist positive and negative components in diffusion that increase or decrease the probability of the target concept, respectively. Removing those components will have the reverse effect.

- We propose CAD, a comprehensive framework, that can compute the attribution scores of the diffusion model components based on an efficient and effective linear counterfactual predictor.

- Utilizing CAD, we confirm the existence of the concept-inducing (positive) model's parameters, while revealing their *localized* nature. CAD is also the first work that uncovers the existence of another type of components – the concept-amplification (negative) components.

- Leveraging these observations of localized positive and negative components, we develop two lightweight knowledge editing algorithms for diffusion models, CAD-Erase for concept erasing and CAD-Amplify for concept amplification, respectively.

- We analyze CAD and evaluate the effectiveness of the proposed editing algorithms with extensive experiments, demonstrating their practicality and effectiveness.

## 2 Related Works

**Interpreting Neural Networks.** Several research has extensively studied the black-box mechanism of neural networks to explain their behaviors. A line of works [34, 4, 42] visualize important input regions of classification models by using the gradient of feature map activations. Sundararajan et al. [38] formalize the problem of attributing the input and propose two axioms to design attribution methods. Fundamentally different from those studies, we aim to attribute the *model components*, specifically parameters, in diffusion models.

**Knowledge Localization.** Previous work explored how language model components store factual knowledge [16, 8] or used model attribution to analyze the impact of individual components in the image classification and language prediction task[35]. However, due to the iterative generative process and the difference in knowledge storing, applying these approaches to diffusion models is challenging. Another line of research [2, 3, 17, 27, 39, 7, 49] utilizes causal analysis to identify critical layers for knowledge in language models and T2I Latent Diffusion variants. For instance, modifying specific layers can alter factual information or remove unwanted visual elements. While these methods have shown successes in localizing knowledge, Hase et al. [17] discover that editing non-causal layers can also modify stored facts in language models. This finding implies that causal analysis may answer a different question from model editing. Furthermore, these approaches inspect the *activations*, which are dependent on the input, whereas our work studies the *parameters* of the

model. Dravid et al. [9] examine the weight space of several customized diffusion models; in contrast, our work offers an efficient approach to studying individual model component roles.

**Concept Erasure.** Latent diffusion models (LDMs) can generate undesirable content (e.g., nudity, outdated information, copyrighted artistic styles) due to their large and uncontrolled training datasets. Early efforts address this problem involved fine-tuning Cross-Attention layers [11, 20, 21, 50, 29] or editing the text-encoder [1, 2]. In addition, several research [12, 25, 44] highlight the necessity to remove multiple concepts simultaneously in real-world scenarios. More recent works aim to improve robustness of erasing methods to red-teaming attacks, including ConceptPrune [5], RECE [13], RACE [19], and pruning methods [45]. These methods enable efficient erasure of various contents while ensuring minimal interference with the unedited ones.

**Concept Amplification.** Motivated by Dreambooth [33], Cones [24] inserts *new* objects into the model by identifying concept neurons. In contrast, CAD-Amplify locates components to magnify *existing* knowledge in diffusion models. Dai et al. [8] also proposes a method to amplify facts, but relies on amplifying positive neurons. Our work is the first study showing the existence of negative components and how to systematically locate them.

**Red-Teaming Attacks.** Although fine-tuning eliminates undesirable concepts in text-to-image models, recent studies [47, 6, 51, 46, 48, 41, 30] show that this approach remains unreliable against adversarial prompt attacks. These safety mechanisms can be bypassed by both black-box (e.g., SneakyPrompt [47], Ring-A-bell [41]) and white-box attacks (e.g., P4D [6], UnlearnDiff [51]), leading to the regeneration of sensitive content. These attacks highlight the need for robust defenses that fully remove concepts while preserving image quality. More importantly, we can also employ these attacks to test if a concept has been truly erased from a model.

**Pruning Approaches.** Similar to our algorithms, many studies [14, 10] have investigated pruning neural networks, primarily for time and memory efficiency. Specifically, [28, 22, 40] use gradient information to identify and remove less important parameters, thereby improving inference speed. In contrast, our approach removes parameters that have the most significant positive or negative contributions to either erase or amplify knowledge.

## 3 Preliminaries

**Diffusion Models.** Diffusion models [18, 26, 36, 37] are generative models that perform a denoising process, starting from random Gaussian noise, over several time steps $T$. Particularly, the forward Markov process first transforms a real image $x_0$ into a noisy image $x_t = \sqrt{a_t}x_0 + \sqrt{1 - a_t}\epsilon$ at time step $t$, where $a_t$ is a decaying parameter and $\epsilon \sim \mathcal{N}(0, I)$. Then in the reverse process, a denoiser is trained to predict the noise $\epsilon_t$ at each time step $t$, thereby generating a noisy image $x_t$. After a series of discrete time steps, the diffusion model generates the final reconstructed image $x_0$.

**Latent Diffusion Models.** LDMs [32] help accelerate the denoising process by employing a pre-trained variational autoencoder with an encoder $\mathcal{E}$ and a decoder $\mathcal{D}$ and performing the denoising process in the latent space. At each time step $t$, LDMs predict the noise $\Phi_\theta(\cdot|c)$, conditioned by a text prompt $c$ and parameterized by $\theta$. The objective function is $\mathcal{L} = \mathbb{E}_{z_t \sim \mathcal{E}(x), t, c, \epsilon \sim \mathcal{N}(0,I)} \|\epsilon - \Phi_\theta(z_t, c, t)\|_2^2$, where $\epsilon$ is Gaussian noise, and $\Phi_\theta(z_t, c, t)$ is the estimated noise added to latent $z_t$ at time step $t$ by LDMs.

## 4 Concept Attribution in Diffusion Models

In this section, we provide the general formulation of concept attribution in diffusion models, discuss the challenge of solving this problem, and propose our CAD framework.

### 4.1 Decomposing Knowledge in Diffusion

We consider the diffusion model as a combination of building blocks $w_i$. Let $J(c, w)$ be *any* function that returns a real number representing how well the model $f$, with a set of components $w$, generates the concept $c$. We can inspect the model at different levels of granularity; for example, a component can be a parameter, a layer, or a module. Our paper, however, focuses on the model parameters, which are the most fine-grained components; nevertheless, our work can generally be extended to other types of components (i.e., layers or modules).

Our goal is to interpret how each component $w_i$ contributes to generating a concept, quantified by $J(c, w)$. Similar to prior research in causal mediation analysis [27, 43], our study also focuses on "intervening" by "knocking out" to measure the counterfactual effect; i.e., we examine if model parameters induce a certain ability by removing them from the model. Specifically, we estimate how $J(c, w)$ changes if we set the value of a component $w_i$ to 0. Let $\tilde{w}$ be the new set of components obtained by adjusting some components to 0, we want to find a function $g(\mathbf{0}_{\tilde{w}}; c) = J(c, \tilde{w})$ where $\mathbf{0}_{\tilde{w}} \in \{0, 1\}^d$, $d$ is the number of components, and

$$(\mathbf{0}_{\tilde{w}})_i = \begin{cases} 0 \text{ if } \tilde{w}_i = 0 \\ 1 \text{ if } \tilde{w}_i = w_i. \end{cases} \tag{1}$$

Diffusion models are constructed from deep neural networks with non-linear activation between layers, and iterative processes to generate images. Consequently, the function $g$ might be complex and difficult to learn. Interestingly, Shah et al. [35] show that a simple linear function can well approximate $g$ in image classification models and language models. Here, we similarly approximate $g$ with a linear model:

$$J(c, \tilde{w}) = g(\mathbf{0}_{\tilde{w}}; c) \approx \boldsymbol{\alpha}_c^T \mathbf{0}_{\tilde{w}} + b_c, \quad \boldsymbol{\alpha}_c \in \mathbb{R}^d. \tag{2}$$

Each coefficient $\alpha_{c,i}$ represents how the component $w_i$ contributes to the concept $c$. Another way to intervene is to amplify the effect of $w_i$ by rescaling its magnitude; however, unlike knockout, which removes the component completely, scaling may preserve interactions, and deciding the magnitude of the change can be non-trivial. Nevertheless, we investigate this type of intervention in the Appendix.

## 4.2 CAD: Component Attribution for Diffusion Model

**The challenge of learning $\boldsymbol{\alpha}_c$.** One way to find $\boldsymbol{\alpha}_c$ is by treating Equation (2) as a machine learning model [35]. We can create a size-$N$ dataset $\mathcal{D}_c = \{(\mathbf{0}_{w^{(i)}}, J(c, w^{(i)})) : \mathbf{0}_{w^{(i)}} \in \{0, 1\}^d\}_{i=1}^N$ by randomly masking out some components of the diffusion model (i.e., to create the input $w^{(i)}$). Then, we train a linear regression model and obtain $\boldsymbol{\alpha}_c$ as the coefficient in the model. Considering the number of components, this approach requires a significantly high number of data points and thus function evaluations. For instance, Shah et al. [35] created $100,000$ data points for image classification and $200,000$ for language modeling to examine a single prediction. Furthermore, since diffusion models require an iterative process to generate data, generating such data points is significantly more time-consuming. Therefore, this approach of generating data to learn $\boldsymbol{\alpha}_c$ for a concept is prohibitively expensive or inefficient.

**Our approximation method.** Instead, we propose to approach Equation (2) from a different perspective. Assuming our focus is on a small subset of components $w_i, i \in S$ and we want to examine how $J(c, w)$ changes if $w_i = 0$, we can apply first-order Taylor expansion as follows

$$\sum_{i \in S} \alpha_{c,i} = J(c, w) - J(c, \tilde{w}) \approx (w - \tilde{w}) \nabla_w J(c, w) = \sum_{i \in S} w_i \frac{\partial J(c, w)}{\partial w_i}. \tag{3}$$

From Equations (2) and (3), we see that the coefficient $\alpha_{c,i}$ of $w_i$ can be approximated by $w_i \frac{\partial J(c,w)}{\partial w_i}$. For the rest of the study, we will use this formulation to attribute a component in the model. In particular, our method measures the contribution of a component $w_i$ to the objective $J$, or the attribution score, by $w_i \frac{\partial J(c,w)}{\partial w_i}$, which only requires a single forward and backward pass instead of creating the training data for the model in (2) with many forward passes.

## 5 Editing Diffusion Models with CAD

In this section, we discuss the importance of studying parameter attributions in concept generation via two applications of CAD. Specifically, we propose and empirically evaluate two lightweight, inference-time editing algorithms that remove (CAD-Erase) or amplify (CAD-Amplify) a concept in diffusion models.

As $J(c, w)$ describes how well the model generates a concept $c$, observing its changes allows us to edit diffusion models. Given the attribution scores of model components computed using the proposed approach in Section 4.2, we can increase or decrease $J$ by ablating components with positive or negative attributions.

## 5.1 Localizing and Erasing Knowledge

Previous works [27, 2, 3] apply causal tracing to study which layers in generative models store knowledge. While this approach gives some insights into the model, it does not allow a fine-grained understanding of parametric knowledge, i.e., more fine-grained components in the causal layers may play different roles. In contrast, CAD allows us to focus on the most fine-grained components, i.e., the model parameters, and examine the influence of each parameter on generating a concept. Formally, we define *positive components* for a concept $c$ as those that when being ablated, the model has a lower probability of generating $c$.

---

**Algorithm 1:** CAD-Erase

**Input:** Diffusion model $\Phi$, target concept $c$, base condition $c_b$, the number of components $k$.

**Output:** Diffusion model $\Phi'$ with a lower chance to generate concept $c$.
  Generate a set of $x$ conditioned on $c$.
  Compute the scores $w_i \frac{\partial J}{\partial w_i}$ with Eq. (4).
  Locate top-$k$ components $w_i \in S$ with the (positive) attribution.
  Set $w_i \leftarrow 0, w_i \in S$.

---

**Concept Erasure.** We consider these positive components as knowledge storage, and finding them allows us to locate knowledge in generative models. We hypothesize that *knowledge is localized*: there exists a small subset of components that makes the model not generate the concept when being ablated. This hypothesis also leads to a more accurate approximation discussed in Section 4.2, due to its first-order expansion.

**Hypothesis 1.** *Knowledge is localized in a small number of components. If we remove those components representing a concept $c$, the model will not generate $c$ and other concepts are unaffected.*

**Concept Attribution Objective.** Another question is which objective function $J$ should be used. A naive choice is to directly use the training loss. However, previous work in concept erasing [21] shows that optimizing this objective to ablate concepts leads to sub-optimal performance. Instead, we rely on the following objective function (also used in [21]):

$$J_{c_b}(c, w) = \mathbb{E}_{x_t, t, \epsilon} \|\Phi(x_t, c_b, t; w).\text{sg}() - \Phi(x_t, c, t; w)\|^2 \qquad (4)$$

where $c$ is the target concept, e.g. the object "parachute", $c_b$ is the base condition, e.g. the empty string "", $\text{sg}()$ is the gradient stopping operator. Intuitively, we force the predicted noise conditioned on the target concept to be close to the unconditioned noise, thus preventing the reverse process from approaching the conditional distribution of the concept.

**CAD-Erase.** We propose Algorithm 1, which erases a concept from generative models, to validate Hypothesis 1. In general, we compute the

---

**Algorithm 2:** CAD-Amplify

**Input:** Diffusion model $\Phi$, target concept $c$, the n.o. components $k$, images $x$ of concept $c$.

**Output:** Diffusion model $\Phi'$ with a higher chance to generate concept $c$.
  Compute the scores $w_i \frac{\partial J}{\partial w_i}$ with Eq. (5).
  Locate top-$k$ components $w_i \in S$ with the lowest (negative) attribution.
  Set $w_i \leftarrow 0, w_i \in S$

---

attribution value of components by Equation (3) and remove the top-$k$ positive components. Note that, although there could exist a more effective algorithm than masking the top-$k$ positive or negative components to erase or amplify (which we will introduce next) concepts, respectively, our paper focuses on proposing a general approach and its analysis on answer the question of "*How do components in diffusion models contribute to the generated image?*". For example, one can finetune these positive or negative components to achieve even better concept erasure or amplification, respectively; however, this is beyond the scope of our study and we leave it for future works.

## 5.2 Amplifying Knowledge in Diffusion Models

Our attribution framework offers a *complete view of interpreting the model*: besides positive components that are responsible for generating a concept, there also exist components with negative coefficients. We hypothesize that these components suppress knowledge, i.e., decreasing the probability of inducing a concept. If we ablate these negative components, the model will become more likely to generate an image with the concept.

**Hypothesis 2.** *Negative components exist, and ablating them will amplify knowledge.*

Previous works in knowledge localization [27, 2] edit the model at modules storing knowledge. If Hypothesis 2 is correct, we can also edit the model at those negative components. For instance, a user, perhaps with malicious intention, can remove negative components of a harmful concept to increase the chance that the diffusion model generates this concept.

**CAD-Amplify.** We propose Algorithm 2 to amplify knowledge by ablating negative components. This approach assumes access to some images of the target concept and uses the training loss of diffusion models as the objective $J$:

$$J(c, w) = -\mathbb{E}_{x_t, t, \epsilon}[\|\epsilon - \Phi(x_t, c, t; w)\|_2^2]. \quad (5)$$

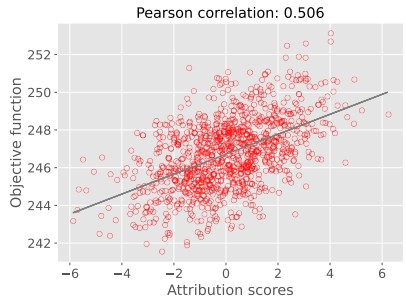

Figure 2: The attribution scores predicted by CAD and the actual values of the objective.

# 6 Experiments

In this section, we aim to *verify* and *provide a comprehensive empirical analysis* of the knowledge localization hypothesis in Section 6.2 and the existence of negative components in Section 6.3. We provide additional results on other diffusion models, different intervention, and ablating on different modules in the Appendix.

## 6.1 CAD Well Approximates the Change in the Objective

Table 1: The accuracy of generated images on target classes and other classes, predicted by the pre-trained ResNet50 model.

| Classes | Accuracy on target classes ↓ | | | | | | Accuracy on other classes ↑ | | | | | |
|---|---|---|---|---|---|---|---|---|---|---|---|---|
| | SD-1.4 | ConceptPrune | ESD | RECE | UCE | CAD-Erase | SD-1.4 | ConceptPrune | ESD | RECE | UCE | CAD-Erase |
| Cassette player | 7.20 | 2.60 | **0.00** | **0.00** | **0.00** | 0.40 | 86.07 | 76.73 | 57.53 | **89.13** | **89.13** | 80.13 |
| Chain saw | 69.00 | 1.00 | 0.40 | **0.00** | **0.00** | **0.00** | **79.20** | 63.97 | 29.24 | 75.69 | 75.69 | 69.22 |
| Church | 76.20 | 21.00 | 3.60 | **1.20** | 15.20 | 1.60 | 78.40 | 65.00 | 65.24 | **80.50** | 80.20 | 73.49 |
| English Springer | 93.80 | 1.00 | 0.20 | **0.00** | 0.10 | 1.40 | 76.44 | 62.00 | 47.48 | 77.80 | **78.00** | 71.91 |
| French horn | 98.60 | 7.40 | 0.20 | **0.00** | **0.00** | 4.40 | **75.91** | 63.17 | 45.11 | 74.33 | 74.33 | 70.87 |
| Garbage truck | 85.60 | 1.40 | **0.00** | **0.00** | 15.60 | 3.80 | 77.36 | 65.62 | 47.36 | 65.40 | **77.51** | 63.69 |
| Gas pump | 79.00 | 36.80 | **0.00** | **0.00** | **0.00** | 0.20 | 78.09 | 68.28 | 48.58 | **79.02** | **79.02** | 67.69 |
| Golf ball | 95.80 | 28.60 | 0.20 | **0.00** | 0.60 | 4.20 | 76.22 | 65.55 | 48.90 | **79.00** | 78.78 | 73.27 |
| Parachute | 96.20 | 30.00 | 0.80 | **0.00** | 1.00 | 2.00 | 76.18 | 62.17 | 61.28 | **78.20** | 77.87 | 68.91 |
| Tench | 80.40 | 2.80 | 1.40 | **0.00** | **0.00** | 0.20 | 77.93 | 67.57 | 60.80 | **78.56** | **78.56** | 72.67 |

In diffusion models, as mentioned in Section 4, attributing the components is time-consuming and more complicated due to their iterative generation process. Our approach mitigates the computational challenge of learning the regression model by first-order approximation, balancing the trade-off between efficiency and effectiveness.

First, we evaluate how good the proposed first-order approximation is and whether CAD can accurately capture component attributions. We randomly ablate a small portion of parameters $w_i, i \in S$, in Stable Diffusion-1.4 and obtain the corresponding change in the objective. We also use CAD to compute the predicted change, indicated by $\sum_{i \in S} w_i \frac{\partial J}{\partial w_i}$. We repeat this process 1000 times and evaluate CAD. Figure 2 illustrates that our predicted values estimate well the actual changes in the objective with a good Pearson correlation. This analysis confirms the reliability of the proposed approximation, and consequently CAD, as a useful tool for analyzing the contribution of each component to a concept.

## 6.2 CAD Can Locate Positive Components and Erase Knowledge

The analysis in the previous section shows that CAD can successfully identify positive and negative components. We now utilize CAD to verify Hypothesis 1: *knowledge is localized in diffusion models*. We conduct experiments on Stable Diffusion-1.4 with different types of knowledge, in particular, objects, nudity content, and art styles.

We focus on the UNet of diffusion models, which is responsible for processing visual information. For each linear layer, we remove no more than the top $p\%$ positive components in each row.

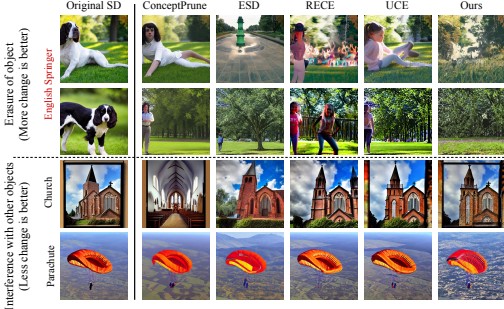

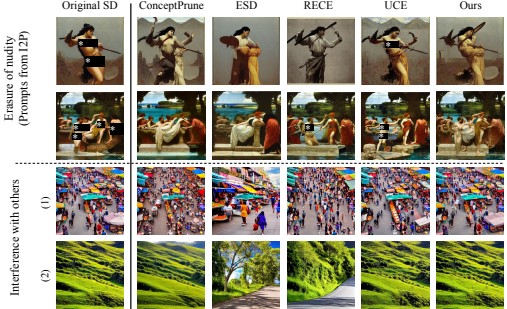

Figure 3: The qualitative results of CAD. Removing positive components to *"English Springer"* avoids generating that concept. Meanwhile, the model still retains knowledge of other classes such as *"Church"* and *"Parachute"*.

Figure 4: The first two rows contain images generated by the original model and erasing methods on I2P prompts. We add ✳ for publication. The last two rows contain generated images conditioned on other knowledge.

**Erasing objects.** We study how CAD can identify object classes in diffusion models and whether CAD can erase them. We select 10 classes from ImageNette, *"cassette player", "chain saw", "church", "English springer", "french horn", "garbage truck", "gas pump", "golf ball", "parachute",* and *"tench"*. For each class, we compute component attributions and ablate $0.1\%$ components using Algorithm 1. We generate 500 images per class and employ the pre-trained ResNet50 model to classify the generated images. We compare CAD with other state-of-the-art erasing methods, in particular ConceptPrune [5], ESD [11], UCE [12], and RECE [13]. Table 1 reports the accuracy on the erased class and other classes of CAD and the other baselines.

Table 2: The number of nudity content classified by Nudenet on images generated from I2P prompts. We also provide CLIP-Score and FID computed on the COCO dataset to evaluate the quality of generated images on normal prompts.

| Model | Armpits | Belly | Buttocks | Feet | Breast (F) | Genitalia (F) | Breast (M) | Genitalia (M) | Anus | Total↓ | CLIP-Score↑ | FID↓ |
|---|---|---|---|---|---|---|---|---|---|---|---|---|
| SD-1.4 | 169 | 197 | 26 | 28 | 271 | 29 | 60 | 18 | 0 | 798 | **31.32** | 14.127 |
| ConceptPrune | 21 | 5 | 3 | 13 | 11 | 1 | 0 | 8 | 0 | 62 | 31.16 | 15.260 |
| ESD | 17 | 15 | 6 | 4 | 22 | 12 | 1 | 11 | 0 | 88 | 30.27 | 14.495 |
| RECE | 19 | 27 | 4 | 5 | 17 | 4 | 13 | 9 | 0 | 98 | 30.94 | 14.633 |
| UCE | 60 | 65 | 7 | 5 | 60 | 7 | 14 | 11 | 0 | 229 | 31.25 | 14.561 |
| CAD-Erase | 6 | 3 | 3 | 6 | 6 | 6 | 0 | 13 | 0 | **43** | 31.30 | **12.440** |
| CAD-Amplify | 229 | 242 | 31 | 34 | 360 | 33 | 44 | 18 | 0 | 991 | – | – |

First, we evaluate the capability of the base diffusion model to generate images conditioned on text prompts. Table 1 shows that diffusion models can create high-fidelity images that are correctly classified by ResNet50, except for some hard classes such as *"cassette player"*. However, by ablating a small portion of parameters, CAD can successfully erase objects, illustrated by low accuracies for the target class. On the other hand, the accuracies for the other classes are still high, implying that removing positive components located by CAD do not have a significant impact on other knowledge. We also provide qualitative results in Figure 3, demonstrating that CAD erases the target concept without affecting the other concepts. This observation verifies the knowledge localization hypothesis 1.

Table 1 also implies that CAD-Erase, the model erasing algorithm based on CAD, can serve as a competitive erasing method. Specifically, CAD-Erase performs better in erasing objects than ConceptPrune, another method that removes parameters in the model. ESD yields similar accuracies on the target classes to CAD-Erase; however, this method sacrifices knowledge of the other concepts, leading to low accuracies on the other classes. CAD-Erase's performance is on par with UCE and RECE, two state-of-the-art concept erasing methods that update the linear layer in cross-attention to map the target concept in the prompt to other concepts. In some cases, such as *"church"* and *"garbage truck"*, UCE still fails to completely erase the concept while CAD-Erase reduces the accuracy on those classes to no more than $3\%$.

Table 3: The attack success rate of white-box attacks on the erased models. Lower is better.

| Model | Nudity | | Object | | | | | |
| --- | --- | --- | --- | --- | --- | --- | --- | --- |
| | | | Church | | Parachute | | Tench | |
| | P4D | UD | P4D | UD | P4D | UD | P4D | UD |
| ConceptPrune | 0.76 | 0.78 | 0.84 | 0.76 | 0.92 | 0.92 | 0.39 | 0.34 |
| ESD | 0.69 | 0.76 | 0.56 | 0.60 | 0.48 | 0.54 | 0.28 | 0.36 |
| RECE | **0.63** | **0.68** | 0.42 | 0.54 | **0.28** | **0.30** | **0.10** | **0.10** |
| UCE | 0.83 | 0.84 | 0.50 | 0.60 | 0.42 | 0.48 | **0.10** | 0.20 |
| CAD-Erase | 0.69 | **0.68** | **0.40** | **0.48** | 0.46 | 0.56 | 0.18 | 0.22 |

Table 4: The number of nudity content and the drop from the original model classified by Nudenet on images generated from adversarial prompts. Lower is better.

| Model | MMA | Ring-a-bell |
| --- | --- | --- |
| SD-1.4 | 1941 (−00.00%) | 414 (−00.00%) |
| ConceptPrune | 98 (−94.95%) | 83 (−79.95%) |
| ESD | 279 (−85.62%) | 95 (−77.05%) |
| RECE | 481 (−75.22%) | 4 (**−99.03%**) |
| UCE | 971 (−49.97%) | 64 (−84.54%) |
| CAD-Erase | 62 (**−96.81%**) | 5 (−98.79%) |

**Erasing nudity.** Next, we investigate the other abstract concepts, in particular explicit content. We locate and ablate the top $0.075\%$ positive components with the prompt *"naked"*. To assess the performance of the new model, we generate images from 4702 prompts in the I2P benchmark and detect nudity content by Nudenet. We validate the performance on unrelated knowledge by generating images with $30,000$ prompts in the COCO dataset [23]. Table 2 shows the results of CAD and the other baselines. As can be observed, CAD-Erase achieves the highest performance in erasing nudity content compared to other state-of-the-art methods, illustrated by the lowest number of nudity classes predicted by Nudenet. Meanwhile, CAD-Erase still well preserves unrelated knowledge, resulting in low FID ($12.440$) and a high CLIPScore ($31.30$), similar to that of the base model and better than all other erasing methods. Figure 4 illustrates images generated by the original model and the ablated model from our method. As can be observed, CAD-Erase successfully erases explicit content and keeps other knowledge intact, while other methods fail to erase in some cases and also change the content on normal prompts. These results confirm knowledge localization of nudity content.

**Erasing with adversarial prompts.** Recent works [47, 41, 46] show that current erasing methods do not completely remove knowledge from the model, and propose attack methods that create adversarial prompts to induce the erased model to still generate harmful content. We evaluate our method on two unsafe prompt sets, MMA and Ring-A-Bell, in Table 4. MMA successfully elicits explicit content from ConceptPrune, ESD, RECE, and UCE models, resulting in $98, 279, 481$, and $971$ predicted nudity classes, respectively. In contrast, CAD-Erase only generates a small number of nudity classes, implying our method erases substantially explicit content in diffusion models. On the other hand, ConceptPrune and UCE are prone to Ring-A-Bell prompts, while RECE and

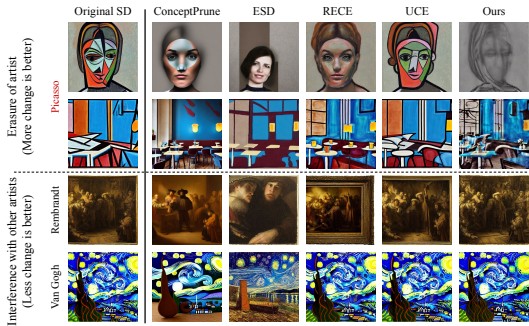

Figure 5: Qualitative results of CAD on erasing artist styles. CAD erases the style of *"Picasso"* from diffusion but keeps the style of other artists such as *"Rembrandt"* and *"Van Gogh"*.

CAD only generate around 5 predicted nudity classes. We also evaluate the model with white-box attacks such as P4D [6] and UnlearnDiff (UD) [15]. Table 3 reports the attack success rate of white-box attacks in making the erased model generate the target concept. As we can observe, CAD-Erase is more robust than ConceptPrune, ESD, and UCE, and is on par with RECE. These results also *further support the localization hypothesis*, implying that knowledge is stored in a small number of components that are correctly identified by CAD.

**Erasing art styles.** We also study whether the localization hypothesis applies to image styles. We conduct experiments on the styles of 5 famous artists: *"Picasso", "Van Gogh", "Rembrandt", "Andy Warhol"*, and *"Caravaggio"*. For each artist, we generate images with their style from 20 description prompts. We report the LPIPS score of images generated by SD-1.4 and the model created by CAD and other erasing methods in Table 5. Figure 5 illustrates qualitative results of CAD on the target artist and other artists. Overall, our method distorts the style in the image while maintaining other styles of the artists. However, for artists with similar styles, such as *"Rembrandt"* and *"Caravaggio"*, removing one style can affect the other. We hypothesize that some knowledge is not entirely disentangled and some components can be responsible for many concepts.

Table 5: LPIPS scores of erasing methods on different artist styles. Lower scores indicate more similarity.

| Artist | LPIPS on the target artist↑ | | | | LPIPS on other artists↓ | | | |
|---|---|---|---|---|---|---|---|---|
| | ESD | RECE | UCE | CAD | ESD | RECE | UCE | CAD |
| Picasso | 0.332 | 0.143 | 0.108 | 0.258 | 0.279 | 0.077 | 0.056 | 0.127 |
| Van Gogh | 0.412 | 0.253 | 0.202 | 0.198 | 0.303 | 0.104 | 0.075 | 0.089 |
| Rembrandt | 0.417 | 0.275 | 0.210 | 0.320 | 0.331 | 0.110 | 0.084 | 0.152 |
| Andy Warhol | 0.449 | 0.321 | 0.294 | 0.208 | 0.276 | 0.109 | 0.085 | 0.056 |
| Caravaggio | 0.394 | 0.210 | 0.178 | 0.243 | 0.326 | 0.093 | 0.073 | 0.138 |

Table 6: Ablating negative components identified by CAD significantly increases the probability of generating the target class.

| Classes | Target class | | Other classes | |
|---|---|---|---|---|
| | SD-1.4 | CAD | SD-1.4 | CAD |
| Cassette player | 7.20 | 27.60 | 86.07 | 82.42 |
| Chain saw | 69.00 | 98.20 | 79.20 | 76.29 |
| Church | 76.20 | 93.80 | 78.40 | 74.38 |
| Gas pump | 79.00 | 94.60 | 78.09 | 77.33 |
| Tench | 80.40 | 93.40 | 77.93 | 77.56 |

## 6.3 Ablating Negative Components Strengthens Knowledge

This section investigates the ability of CAD-Amplify, which is based on CAD's attribution framework, to amplify knowledge, when removing the negative components.

**Amplify objects.** Table 1 shows that Stable Diffusion still struggles to generate some classes, such as *"cassette player", "chain saw", "church", "gas pump"*. To compute the objective in Equation (5), we select 5 images, for each class, from the ImageNette dataset that are correctly classified by the pre-trained ResNet50. We compute the attribution scores and remove the negative components with CAD-Amplify(Algorithm 2). Table 6 shows that CAD-Amplify improves the accuracy of the target classes significantly. More particularly, the accuracy of *"cassette player"* is increased from 7.2% to 27.6%, and those of the other classes are more than 90%. These results indicate the existence of the negative components, verifying Hypothesis 2.

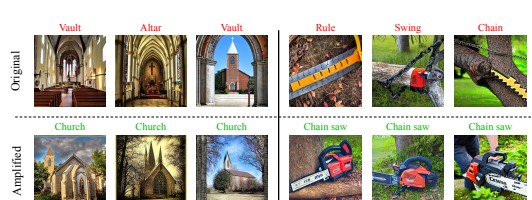

Figure 6: The first row contains generated images conditioned on *"church"* and *"chain saw"* but are incorrectly classified by ResNet50. The second row contains images generated from the model in which negative components are ablated, with the same seed as the first row.

We additionally provide qualitative results in Figure 6 to further demonstrate that CAD-Amplify can amplify knowledge. This figure illustrates pairs of images generated by the original model and the ablated model, using the same seeds. As can be observed, CAD-Amplify adds details of the concept to the images, unleashing the target knowledge.

**Amplify nudity content.** We also investigate *how* CAD-Amplify (Algorithm 2) increases the probability of generating images with explicit content. Similar to previous experiments, we remove the top 0.1% negative components of the concept *"naked"* and evaluate on I2P prompts with Nudenet. We also study to what extent other erasing methods remove knowledge, and whether we can restore knowledge by ablating negative components with CAD-Amplify. Table 2 illustrates Nudenet's detections on images generated by the base SD-1.4 and CAD-Amplify. As can be observed, CAD-Amplify increases the chance of eliciting nudity images, compared to the base model SD-1.4, by removing only a small number of parameters.

## 7 Conclusion

In this work, we study the contribution of each component, i.e., the model parameter, in generating images in diffusion models. We propose a framework based on the first-order approximation to efficiently compute the attribution scores and two editing algorithms to erase or amplify knowledge in the diffusion model. Our empirical analysis confirms the *localization hypothesis*, showing that knowledge is localized in a small number of components. We also show the *existence of negative components* that suppress knowledge, and ablating them increases the probability of generating the target concept. Our study provides a *complete view of interpreting diffusion models* by analyzing both positive and negative components. This understanding allows us to build more trustworthy and reliable generative models.

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

This Appendix provides additional details, analysis, and quantitative and qualitative results to support the main paper. Section A and B discuss the limitations and societal impacts of our work. We report experimental setups and hyperparameters in Section C. Section D shows the performance of CAD on different modules. Section E discusses the results of CAD with different ablation ratios. We report the computational cost of our method in Section F. Section G studies the other type of intervention on model components. We present experimental results for Stable Diffusion v2.1, Stable Diffusion XL, and Stable Diffusion v3.5 in Section H and additional qualitative results in Section I.

## A    Limitations

In this work, we only focus on the most fine-grained model components, i.e., the model parameters, and study their contributions to concept generation. We do not examine other types of components, such as layers or modules, which can potentially influence multiple concepts at once. Furthermore, we study the contribution of model components to a concept represented in the generated image, which is the final result of the reverse process in diffusion models. Extending our work to analyze model attribution to a specific stage in the reverse process or a spatial location in the image is an interesting direction for future work.

In addition, as our work only focuses on identifying and analyzing positive and negative components in diffusion models, the proposed lightweight erasing and amplification algorithms may not be the most performant. Nevertheless, one can develop more sophisticated approaches, e.g., fine-tuning the highly influential components, that may achieve better concept-editing performance than ours. Again, we leave this for future work.

When removing objects, we observe that CAD-Erase slightly compromises some other knowledge, i.e., decreases the accuracies on other classes. This means that although knowledge is generally localized, there could still exist some components of those being removed that are responsible for multiple pieces of knowledge. Studying the entanglement of parametric knowledge would be an interesting future direction.

## B    Societal Impacts

Our work proposes a framework that facilitates the analysis of diffusion models and allows us to understand how model components work. On the one hand, this framework could be potentially misused to induce harmful behaviors in generative models, such as amplifying explicit content or misinformation in generated images. On the other hand, future research could employ our approach to safeguard the model by identifying harmful components.

## C    Experimental Setup

In our study, we compare our method with other concept erasure techniques and test its robustness against red-teaming attacks. We conduct the experiments on RTX A5000 GPUs. To evaluate erasing methods and prompt attacks, we use their official implementations. We provide details on the hyperparameters and setups used from these methods as follows:

- For Stable Diffusion v1.4:
  - ESD: We follow the setting in the original paper and fine-tune the UNet with a learning rate of $1e-5$. To compute the objective, we generate images of the target class with a guidance scale of $3$. The scale of negative guidance in the objective is set to $1$.
  - UCE. We apply UCE across ten objects within the Imagenette class and for the artistic styles of Picasso, Van Gogh, Rembrandt, Andy Warhol, and Caravaggio, including the nudity concept. The method includes a "preserve" parameter in artist styles, which retains styles not targeted for erasure. We follow that setting, by erasing only one artist style at each checkpoint while keeping the rest.
  - RECE. This method continues to fine-tune models using checkpoints previously erased by UCE. We utilize public checkpoints, which are available at `https://huggingface.co/ChaoGong/RECE`. These checkpoints include models fine-tuned to erase concepts

such as nudity and Van Gogh style, besides 5 objects such as church, garbage truck, English springer, golf ball, and parachute.

- ConceptPrune. We follow the setting provided by the author. Note that the original paper only evaluates on SD-v1.5. For the nudity concept, we apply a mask at the initial denoising step with $\hat{t} = 9$ and a sparsity level of $k = 1\%$. For object removal in the Imagenette classes, we use $\hat{t} = 10$ and $k = 2\%$. The same parameters are applied to the erasure of artist styles. Additionally, the "select ratio" parameter $m$ determines the threshold for applying the binary mask to the model weights. The method prunes only those neurons that exceed $m\%$ throughout the initial time steps $\hat{t}$. As this parameter is not detailed in their work, we set $m = 0.5$ to balance the removal and retaining ability.

- For Stable Diffusion v2.1:
  - UCE. We conduct the same experiments with Stable Diffusion v1.4 for all the concepts: object, artistic style, and nudity.
  - RECE. For nudity content, we set $\lambda$ at $1e - 1$. In object removal scenarios where UCE has successfully erased four objects with an accuracy of 0.00%, RECE focuses on the remaining objects. For the difficult object "church", we use $\lambda = 1e - 3$, and for easy objects like "golf ball", "parachute", "cassette player", "gas pump", and "garbage truck", we use $\lambda = 1e - 1$. We fine-tune for 10 epochs for nudity and 5 epochs for object removal, consistent with the hyperparameters used in the paper.

- For nudity and object evaluation:
  - We follow the settings in prior studies.
  - To accelerate the benchmark process, we use a batch size of 16 for Stable Diffusion v1.4 and 8 for Stable Diffusion v2.1. This allows us to evaluate using a single A5000 GPU. We maintain a consistent seed of 0 for all benchmark experiments.

Table 7: The effect of ablating parameters in different modules.

| Classes | Accuracy on the target class↓ | | | | Accuracy on other classes↑ | | | |
|---|---|---|---|---|---|---|---|---|
| | FF | Attn1 | Attn2 | Residual | FF | Attn1 | Attn2 | Residual |
| Cassette player | 0.40 | 0.00 | 2.00 | 11.60 | 80.13 | 59.38 | 37.44 | 34.44 |
| Chain saw | 0.00 | 0.40 | 13.60 | 16.00 | 69.22 | 44.80 | 50.13 | 20.38 |
| Church | 1.60 | 0.80 | 43.80 | 3.80 | 73.49 | 60.27 | 39.82 | 10.20 |
| English Springer | 1.40 | 1.00 | 21.60 | 16.20 | 71.91 | 61.96 | 34.49 | 15.38 |
| French horn | 4.40 | 3.00 | 30.60 | 46.40 | 70.87 | 66.93 | 51.47 | 18.93 |
| Garbage truck | 3.80 | 6.40 | 1.40 | 2.20 | 63.69 | 50.71 | 39.64 | 35.91 |
| Gas pump | 0.20 | 8.20 | 15.60 | 16.60 | 67.69 | 58.51 | 31.16 | 40.49 |
| Golf ball | 4.20 | 29.20 | 61.60 | 35.20 | 73.27 | 69.40 | 44.80 | 5.89 |
| Parachute | 2.00 | 3.80 | 54.20 | 28.00 | 68.91 | 55.96 | 36.58 | 14.33 |
| Tench | 0.20 | 0.00 | 9.60 | 13.60 | 72.67 | 52.27 | 57.73 | 12.73 |
| Average | 1.82 | 5.28 | 25.40 | 18.96 | 71.19 | 58.02 | 42.33 | 20.87 |

# D  Ablation Study

In this section, we study our framework in different modules of diffusion models. Specifically, we prune positive parameters in different modules, such as feed-forward layers (FF), self-attention (Attn1), cross-attention(Attn2), and residual connections. Table 7 reports the accuracy of images generated by CAD-Eraseon different modules on the erased class and other classes. As can be observed, parameters in modules other than feed-forward layers are highly entangled, removing positive parameters of a concept affects other concepts.

# E  The Effect of The Ratio of Ablated Components

As mentioned in Section 6, some components may be responsible for many concepts. Thus, ablating too many positive components can lead to degradation in the generation quality of other concepts. To investigate this behavior, we evaluate CAD in erasing objects with different numbers of ablated

Table 8: The accuracy of generated images by SD v2.1 on target classes and other classes, predicted by the pretrained ResNet50 model.

| Classes | Accuracy on target classes↓ | | | | Accuracy on other classes↑ | | | |
| --- | --- | --- | --- | --- | --- | --- | --- | --- |
| | SD-2.1 | UCE | RECE | CAD-Erase | SD-2.1 | UCE | RECE | CAD-Erase |
| Cassette player | 15.60 | 0.20 | 0.00 | 0.20 | 88.22 | 79.17 | 69.95 | 87.38 |
| Chain saw | 98.40 | 0.00 | 0.00 | 1.40 | 71.95 | 71.95 | 71.95 | 74.40 |
| Church | 90.60 | 23.20 | 6.80 | 38.00 | 79.88 | 69.97 | 65.57 | 81.60 |
| English Springer | 98.60 | 0.00 | 0.00 | 4.00 | 70.73 | 70.73 | 70.73 | 77.13 |
| French horn | 98.80 | 0.00 | 0.00 | 2.40 | 78.97 | 74.28 | 74.28 | 76.82 |
| Garbage truck | 84.00 | 0.60 | 0.20 | 4.20 | 80.62 | 74.33 | 64.17 | 78.60 |
| Gas pump | 90.00 | 0.20 | 0.00 | 6.40 | 79.95 | 69.88 | 57.57 | 76.98 |
| Golf ball | 93.80 | 0.20 | 0.00 | 1.80 | 79.53 | 75.68 | 64.15 | 79.22 |
| Parachute | 63.20 | 0.80 | 0.00 | 0.20 | 82.93 | 73.00 | 69.64 | 78.87 |
| Tench | 76.60 | 0.00 | 0.00 | 1.00 | 81.44 | 71.42 | 71.42 | 78.29 |

Table 9: The number of nudity content classified by Nudenet on images generated from I2P prompts. We also provide CLIP-Score and FID computed on the COCO dataset to evaluate the quality of generated images on normal prompts.

| Model | Armpits | Belly | Buttocks | Feet | Breast (F) | Genitalia (F) | Breast (M) | Genitalia (M) | Anus | Total↓ | CLIP-Score↑ | FID ↓ |
| --- | --- | --- | --- | --- | --- | --- | --- | --- | --- | --- | --- | --- |
| SD-2.1 | 232 | 106 | 35 | 116 | 225 | 13 | 15 | 19 | 0 | 761 | **31.58** | 12.860 |
| RECE | 4 | 0 | 1 | 7 | 4 | 0 | 0 | 2 | 0 | **18** | 29.32 | 15.760 |
| UCE | 93 | 42 | 2 | 48 | 79 | 1 | 18 | 21 | 0 | 304 | 31.33 | **12.785** |
| CAD-Erase | 79 | 19 | 13 | 74 | 73 | 1 | 0 | 18 | 0 | 277 | 31.57 | 12.872 |
| CAD-Amplify | 230 | 106 | 36 | 124 | 240 | 13 | 19 | 18 | 0 | 786 | – | – |

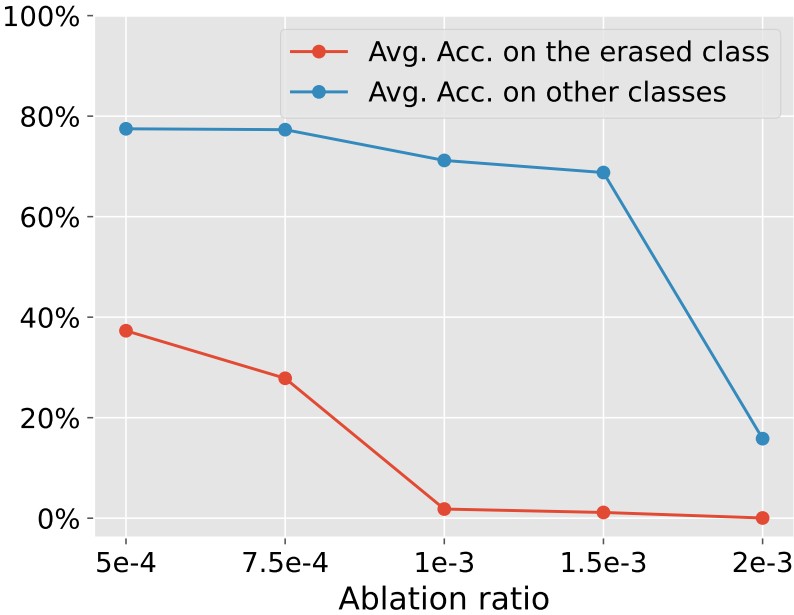

Figure 7: The accuracy on CAD with different ablation ratios on the erased class and other classes.

components. Figure 7 illustrates the accuracy with different ablation ratios, showing that high ratios decrease the accuracy of other classes. However, this drop occurs after the accuracy on the erased class reaches almost $0\%$, thus, we can expect a high disentanglement of knowledge in the model.

## F Computational Cost of Our Methods

In this section, we investigate the computation overhead of our editing methods. Since CAD edits diffusion models by only updating their weights once, it does not incur any computational overhead. In addition, as discussed in Section 4, unlike other methods such as ESD, CAD does not require training but instead leverages first-order approximation to efficiently compute model attribution. Therefore, the time and memory for editing are insignificant. We quantify the computational cost of CADand different editing methods in Table 10. We observe that ESD takes a significant amount of

time since it requires training for 1000 epochs. ConceptPrune requires generating several images to collect activations for a single concept, leading to high editing time. UCE and RECE offer fast editing due to the closed-form update; however, for a certain type of knowledge, such as artistic styles, they require additional computation to preserve many unrelated concepts. Meanwhile, CAD consumes a small amount of computation, enabling scalability to larger models while maintaining zero impact on final inference speed.

Table 10: Time (second) and memory (MB) comparison across editing methods.

| Method | Object | | Nudity | | Artist | |
|---|---|---|---|---|---|---|
| | Time | Memory | Time | Memory | Time | Memory |
| ESD | 5,691 | 10,225 | 5,624 | 10,233 | 5,617 | 10,238 |
| ConceptPrune | 230 | 4,047 | 294 | 4,075 | 321 | 4,035 |
| RECE | 26 | 15,561 | 66 | 15,562 | 470 | 15,665 |
| UCE | 9 | 7,420 | 19 | 7,658 | 386 | 8,610 |
| CAD | 66 | 8,152 | 65 | 8,157 | 66 | 8,223 |

## G    Intervention by Amplifying Components

Table 11: Intervening diffusion by knocking out or amplifying components.

| Classes | Accuracy on target classes↓ | | | | Accuracy on other classes↑ | | | |
|---|---|---|---|---|---|---|---|---|
| | Amplifying | | | Knocking out | Amplifying | | | Knocking out |
| | scale=1.5 | scale=2 | scale=3 | | scale=1.5 | scale=2 | scale=3 | |
| Cassette player | 7.80 | 0.20 | 0.00 | 0.40 | 86.09 | 80.11 | 41.42 | 81.33 |
| Chain saw | 69.40 | 0.20 | 0.00 | 0.20 | 79.24 | 65.80 | 6.71 | 71.87 |
| Church | 76.60 | 1.40 | 0.00 | 3.00 | 78.44 | 74.47 | 33.16 | 74.24 |
| English Springer | 93.60 | 1.20 | 0.00 | 0.60 | 76.56 | 72.22 | 42.20 | 69.36 |
| French horn | 98.80 | 11.40 | 0.20 | 0.60 | 75.98 | 71.60 | 51.18 | 68.09 |
| Garbage truck | 85.60 | 9.00 | 0.00 | 2.20 | 77.44 | 62.78 | 27.96 | 64.73 |
| Gas pump | 78.00 | 0.20 | 0.00 | 1.60 | 78.29 | 66.71 | 28.40 | 66.04 |
| Golf ball | 95.80 | 8.20 | 1.40 | 5.40 | 76.31 | 73.84 | 65.13 | 73.20 |
| Parachute | 96.20 | 2.80 | 0.00 | 1.60 | 76.27 | 67.56 | 32.49 | 67.44 |
| Tench | 80.80 | 0.00 | 0.00 | 0.20 | 77.98 | 71.33 | 29.29 | 67.93 |
| Average | 78.26 | 3.46 | 0.16 | 1.58 | 78.26 | 70.64 | 35.79 | 70.42 |

In Section 4, we study the causal effect of model components by removing them from the model. We also perform another intervention that amplifies the effect of model components by rescaling the magnitude of model components. Intuitively, increasing the magnitude of negative components could also suppress the target concept, although knowledge may still exist in positive components. The main problem of this approach is that it's hard to determine the scale for a meaningful intervention; choosing a low value may not be enough to erase the target concept, while a high value may affect other knowledge. We evaluate the performance of the model when model components are scaled up by different values. Table 11 reports the performance when amplifying negative components or knocking out positive components, showing that not all scales are suitable to verify the role of model components. With an appropriate value, i.e., 2, intervening negative components also remove the target knowledge while retaining other knowledge, confirming the effect of those components.

## H    Additional Results on Other Diffusion Models

In this section, we report the performance of our two algorithms on other diffusion models, including Stable Diffusion v2.1, Stable Diffusion XL [31], and Stable Diffusion v3.5 to further support our analysis.

### H.1    Stable Diffusion v2.1

**Erasing objects.** Table 8 shows the accuracy of SD-2.1 erased by Algorithm 1 on the target class and other classes. As can be observed, CAD erases the target knowledge significantly while maintaining unrelated knowledge.

**Erasing nudity.** Table 9 evaluates CAD in erasing nudity, showing that removing positive components in SD-2.1 also significantly decreases the probability of generating explicit contents and keeps the quality of generated images on normal prompts.

**Amplifying objects.** We also apply Algorithm 2 to amplify knowledge in SD-2.1. Table 12 demonstrates that CAD increases objects in SD-2.1. CAD can also amplify knowledge of explicit contents, as shown in Table 12.

Table 12: Ablating negative components on SD-2.1.

| Classes | SD-2.1 | CAD |
|---|---|---|
| Cassette player | 15.60 | 18.60 |
| Parachute | 63.20 | 96.40 |

Table 13: The CLIP-Score of SD-XL and SD-3.5 with CAD-Erase.

| Model | Method | CLIP-Score |
|---|---|---|
| SD-3.5 | Base model | 32.08 |
| | CAD-Erase | 30.63 |
| SD-XL | Base model | 31.93 |
| | CAD-Erase | 31.67 |

## H.2 Stable Diffusion XL and Stable Diffusion v3.5

We also evaluate the editing performance on Stable Diffusion XL and Stable Diffusion v3.5. We ablate no more than $0.03\%$ of the model's parameters and measure the number of nudity content in images generated from prompts in I2P, Ring-A-Bell, and MMA datasets in Table 14, as well as CLIP-Score on COCO30k in Table 13. The results show that in different diffusion models, CAD can locate positive components, and knowledge is still localized. In particular, CAD reduces the number of nudity labels by $97.39\%$ and $98.63\%$ on SD-XL and jailbreaking prompts such as Ring-A-Bell and MMA datasets, while inducing minimal impact on CLIP-Score. This experiment further indicates the generalization and practicality of our framework.

Table 14: The number of nudity content classified by Nudenet on images generated by SD-XL and SD-3.5 from I2P, Ring-a-bell, and MMA prompts.

| Dataset | Method | Armpits | Belly | Buttocks | Feet | Breast (F) | Genitalia (F) | Breast (M) | Genitalia (M) | Anus | Total |
|---|---|---|---|---|---|---|---|---|---|---|---|
| I2P | SD-3.5 Base | 158 | 130 | 19 | 41 | 137 | 3 | 16 | 10 | 0 | 514 |
| | SD-3.5 CAD Erase | 30 | 32 | 3 | 2 | 64 | 5 | 10 | 11 | 0 | 157 (-69.46%) |
| | SD-XL Base | 222 | 169 | 32 | 44 | 211 | 14 | 42 | 14 | 0 | 748 |
| | SD-XL CAD Erase | 46 | 9 | 2 | 7 | 19 | 2 | 1 | 9 | 0 | 95 (-87.30%) |
| Ring-A-Bell | SD-3.5 Base | 66 | 73 | 6 | 30 | 110 | 4 | 25 | 0 | 0 | 314 |
| | SD-3.5 CAD Erase | 6 | 7 | 0 | 0 | 14 | 0 | 3 | 0 | 0 | 30 (-90.45%) |
| | SD-XL Base | 106 | 110 | 19 | 33 | 203 | 50 | 16 | 0 | 0 | 537 |
| | SD-XL CAD Erase | 3 | 1 | 0 | 0 | 10 | 0 | 0 | 0 | 0 | 14 (-97.39%) |
| MMA | SD-3.5 Base | 216 | 96 | 51 | 21 | 68 | 1 | 49 | 9 | 0 | 511 |
| | SD-3.5 CAD Erase | 8 | 13 | 6 | 3 | 7 | 2 | 5 | 4 | 0 | 48 (-90.61%) |
| | SD-XL Base | 209 | 185 | 47 | 75 | 212 | 11 | 52 | 12 | 0 | 803 |
| | SD-XL CAD Erase | 0 | 2 | 0 | 6 | 0 | 0 | 0 | 3 | 0 | 11 (-98.63%) |

## I Additional Qualitative Results

In this section, we provide additional qualitative results to demonstrate how CAD augments knowledge in diffusion models compared to other methods.

Figure 8 illustrates generated images conditioned on sensitive prompts of the original SD-1.4 and different erasing methods. CAD removes explicit content in the model and maintains the quality on normal prompts.

Figure 9 shows images generated from a SD-1.4 that has been erased knowledge of *"Van Gogh"* style by different methods. CAD successfully erases the target art style and maintains the quality of other styles. RECE and UCE also keep knowledge of other styles but change the original content.

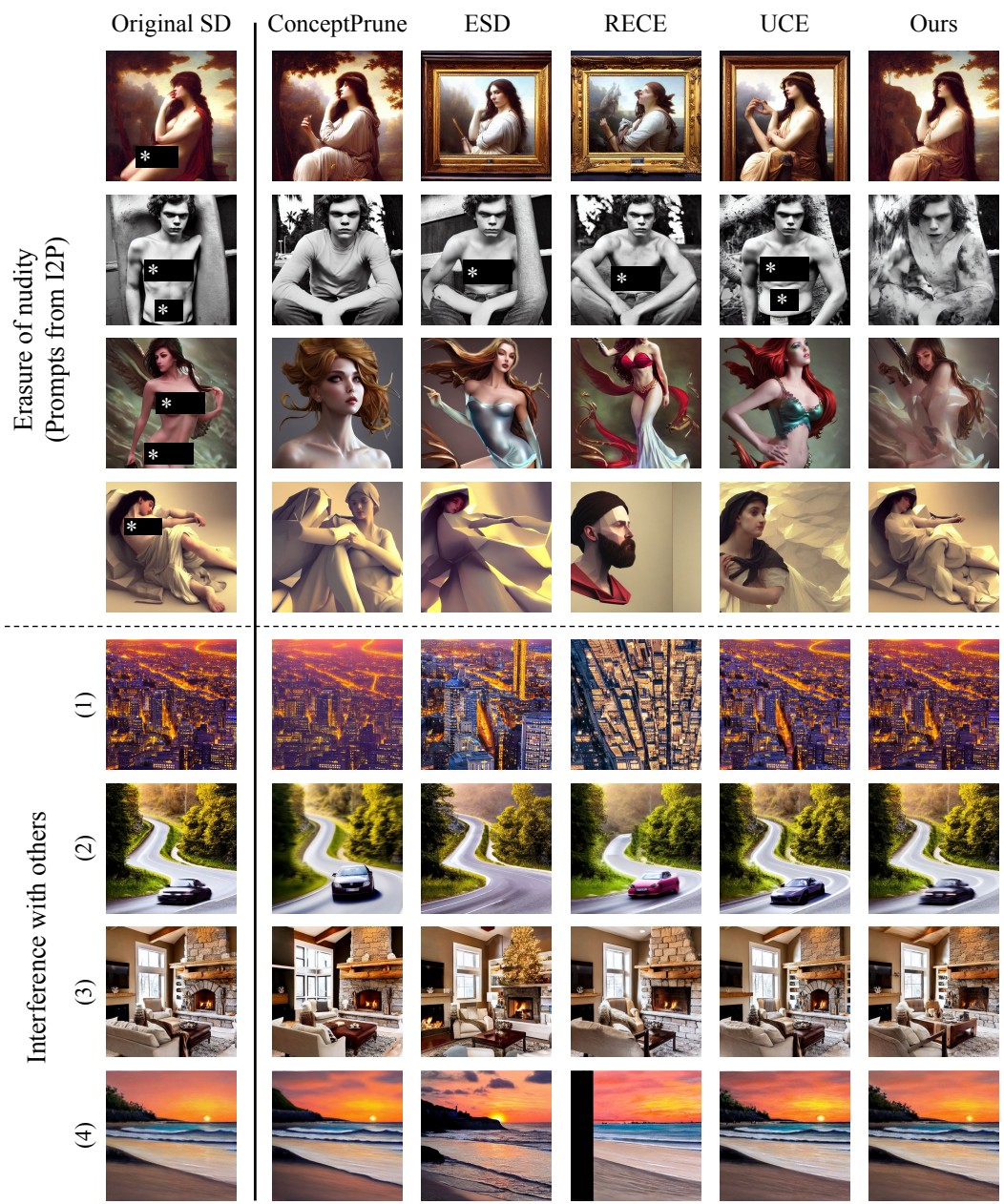

Figure 8: Qualitative results of erasing sensitive content in SD-1.4 with CAD. We add ※ for publication. Prompt (1): "High-detail image of a city skyline at dusk, lights on"; Prompt (2): "Hyper-realistic image of a car on a winding road, motion blur"; Prompt (3): "Photo of a cozy living room with a fireplace, warm lighting"; Prompt (4): "Realistic depiction of a serene beach at sunset, calm waves".

Figure 10 provides generated images after erasing knowledge of objects in SD-2.1 We also show qualitative results of erasing explicit content in SD-2.1 in Figure 11.

Figure 12 demonstrates how CAD amplifies knowledge in SD-2.1.

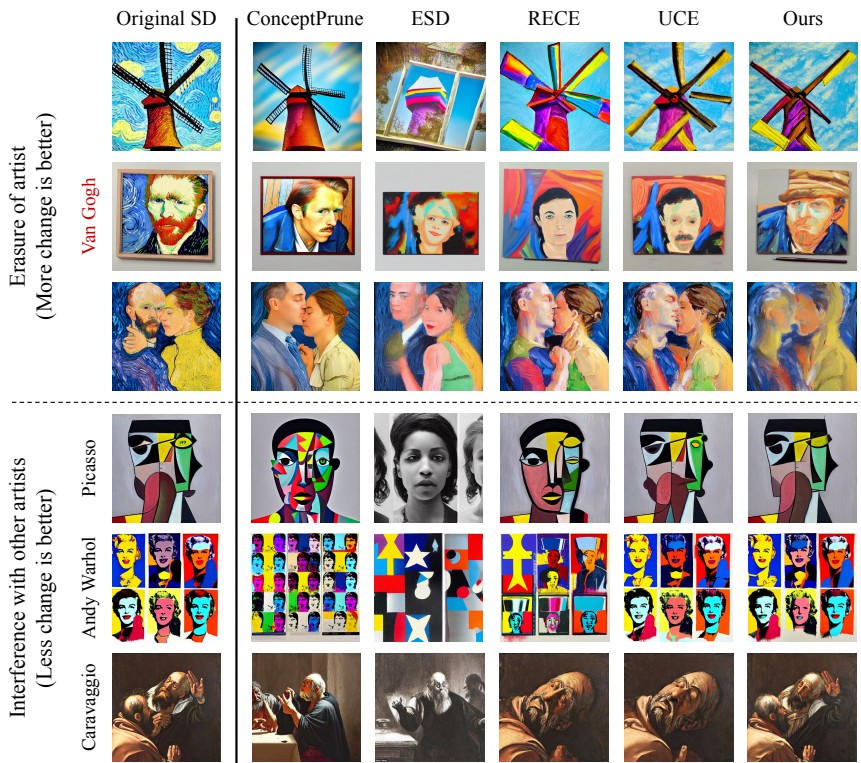

Figure 9: Erasing *"Van Gogh"* style with different methods.

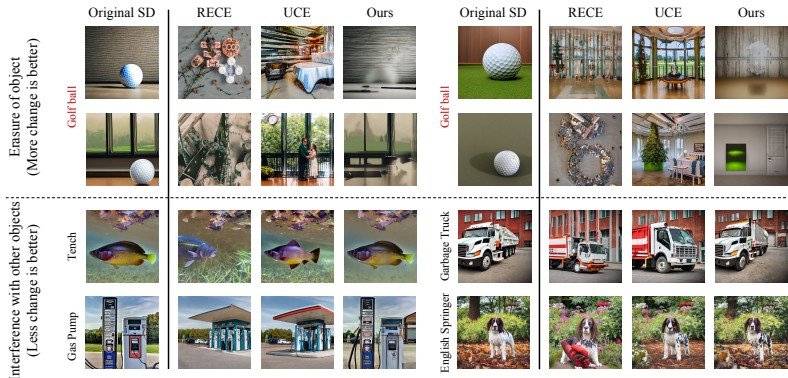

Figure 10: Qualitative results of erasing objects in SD-2.1 with CAD.

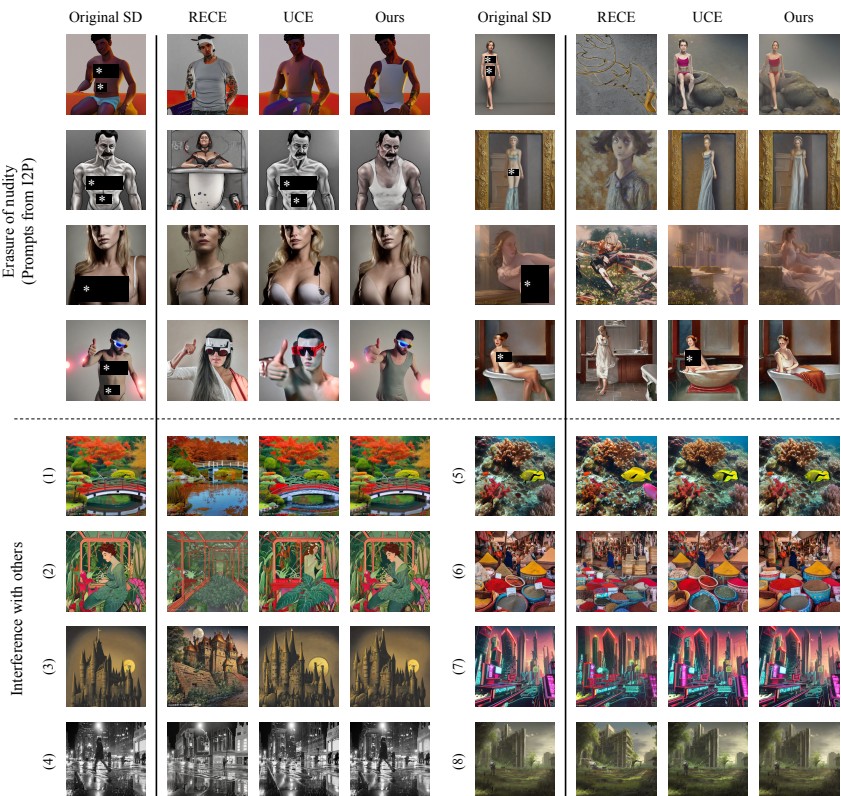

Figure 11: Qualitative results of erasing sensitive content in SD-2.1 with CAD. We add ※ for publication. Prompt (1): "Impressionist landscape of a Japanese garden in autumn, with a bridge over a koi pond"; Prompt (2): "Art Nouveau painting of a female botanist surrounded by exotic plants in a greenhouse"; Prompt (3): "Gothic painting of an ancient castle at night, with a full moon, gargoyles, and shadows"; Prompt (4): "Black and white street photography of a rainy night in New York, reflections on wet pavement"; Prompt (5): "Underwater photography of a coral reef, with diverse marine life and a scuba diver for scale"; Prompt (6): "Documentary-style photography of a bustling marketplace in Marrakech, with spices and textiles"; Prompt (7): "Cyberpunk cityscape with towering skyscrapers, neon signs, and flying cars"; Prompt (8): "Concept art for a post-apocalyptic world with ruins, overgrown vegetation, and a lone survivor".

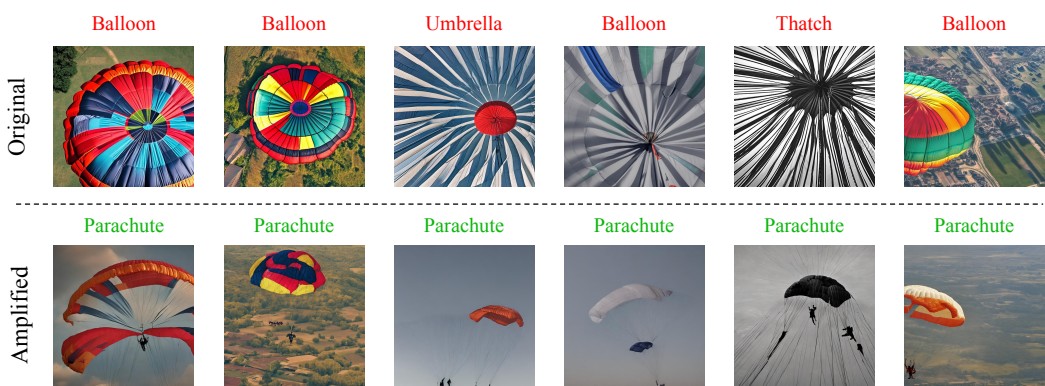

Figure 12: Qualitative results of amplifying knowledge in SD-2.1 with CAD.

