# OpenReview forum: "Unveiling Concept Attribution in Diffusion Models"
_NeurIPS.cc/2025/Conference — NeurIPS 2025 poster_

### Official Review · Reviewer_Yq3z · 2025-06-16

**Clarity:** 3
**Significance:** 2
**Originality:** 2
**Rating:** 2
**Confidence:** 4

**Summary:**

The authors propose a novel framework, Component Attribution for Diffusion Model (CAD), to interpret how individual model parameters in diffusion models contribute to generating specific concepts (e.g., objects, styles, or explicit content). By leveraging a first-order approximation to compute attribution scores, the authors identify both positive (concept-inducing) and negative (concept-suppressing) components. They further introduce two inference-time editing algorithms, CAD-Erase and CAD-Amplify, to respectively remove or enhance specific concepts in generated images.

**Questions:**

While the paper introduces parameter-level attribution and negative components, these contributions are incremental rather than transformative. The framework builds on existing methods like causal tracing [2, 3, 27] and linear counterfactual estimation [34], with the primary novelty being the identification of negative components. However, this finding is underexplored, and the overall approach does not sufficiently distinguish itself from prior interpretability work in generative models. The editing algorithms, while practical, are similar in spirit to existing methods like ConceptPrune and UCE, reducing the paper’s originality.

While the paper claims novelty in identifying negative components and parameter-level attribution, the core idea of attributing model components builds heavily on prior work like Shah et al. [34] and knowledge localization studies [2, 3, 27].

The main score and formulation used (Eq. 4) is not novel [21,27] as acknowledged by the authors.

The discovery of negative components, while interesting, lacks sufficient exploration to establish its significance or differentiate it from existing interpretability frameworks. The paper does not convincingly demonstrate that CAD provides a transformative advance over established methods.

Weak Theoretical Foundation: The reliance on a first-order approximation for attribution scores (Section 4.2) is empirically validated (Figure 2) but lacks theoretical grounding. The paper explicitly notes the absence of theoretical results (Page 13), and the justification for the approximation’s generalizability is insufficient. Without a deeper analysis of its limitations, the method’s reliability across diverse models and concepts remains questionable.

Incomplete Analysis of Negative Components: The identification of negative components is a key claim, but the paper provides minimal insight into their nature, origin, or role in diffusion models. For example, it is unclear whether these components are training artifacts or intentional suppressors, and their distribution across model layers is not explored. This limits the theoretical and practical impact of the finding.

Narrow Experimental Scope: The evaluation is limited to Stable Diffusion-1.4, which raises concerns about generalizability to other diffusion models or newer variants. The paper does not discuss challenges or adaptations needed for applying CAD to other architectures, undermining its broader applicability.

Concept Entanglement Issues: The paper notes that erasing one concept (e.g., Rembrandt’s style) can affect similar concepts (e.g., Caravaggio’s style, Page 8), indicating entanglement in parameter contributions. This issue is acknowledged but not systematically analyzed or addressed, limiting the practical utility of CAD-Erase for complex or overlapping concepts.

Lack of Statistical Rigor: The paper fixes seeds for experiments (Page 15) but does not report error bars, confidence intervals, or statistical significance tests for key metrics (e.g., accuracy, FID, LPIPS). This weakens the reliability of comparisons with baselines and the robustness of the claims.

**Ethical Concerns:**

["NO or VERY MINOR ethics concerns only"]

**Final Justification:**

I appreciate the additional experiments presented in the rebuttal, as well as the inclusion of error bars in the tables. However, I continue to believe that the contribution of this work remains limited in terms of novelty and the overall scope of the effort.

**Limitations:**

Yes

**Quality:**

2

**Strengths And Weaknesses:**

Strengths:

1. Focus on Parameter-Level Analysis: The paper's emphasis on fine-grained, parameter-level attribution is a step beyond layer-based analyses common in prior work. This granular approach could potentially offer deeper insights into diffusion model mechanics.

2. Practical Applications: The proposed CAD-Erase and CAD-Amplify algorithms provide practical tools for concept manipulation, with potential applications in content moderation (e.g., erasing nudity) and enhancing underrepresented concepts.

---

> ### Author Rebuttal · Authors · 2025-07-31
>
> We really appreciate your insightful comments. Please see our response below.
>
> **Q1:** While the paper introduces parameter-level attribution and negative components, these contributions are incremental rather than transformative.
>
> **A:** We'd like to clarify the contributions of our paper compared to the literature. Previous works on localizing knowledge in diffusion [2,3] focus on coarse-grained components such as layers, thus, may over-generalize and misinterpret their roles. For example, [2] concludes that knowledge is distributed amongst UNet and localized in text-encoder. On the other hand, since our framework focuses on fine-grained components, we confirm the localization hypothesis and show that knowledge is localized in a small number of parameters (Section 6.2).
>
> Secondly, although Shah et al. [34] explore the linear counterfactual estimator, they only apply it to classification models. As discussed in Section 4.2, this approach is inefficient and challenging for diffusion models. Our work leverages first-order approximation, showing that it could accurately approximate the objective (Section 6.1) without expensive computational overhead and help us locate positive and negative components.
>
> Finally, the primary goal of our framework is to identify positive and negative components, which is distinctive from methods that erase knowledge from the model, such as ConceptPrune and UCE. In particular, ConceptPrune removes model parameters based on feature norms, while UCE updates the projection matrix with a closed-form solution. In contrast, by computing the attribution score, we show that there are parameters contributing positively or negatively to a knowledge, and ablating them from the model leads to the inverse effect.
>
> **Q2:** the core idea of attributing model components builds heavily on prior work like Shah et al. [34] and knowledge localization studies [2, 3, 27]. The main score and formulation used (Eq. 4) is not novel [21,27] as acknowledged by the authors.
>
> **A:** As discussed in the previous answer, our work offers an advanced methodology compared to [34] and provides novel insights compared to [2,3,27].
>
> Although prior works have explored Eq. 4, they directly fine-tune the model to minimize that objective. In contrast, we show that Eq. 4 could be used to identify positive components, which is interesting and hasn't been observed. This property facilitates the proof of the localization hypothesis, which is the main contribution of our paper.
>
> **Q3:** The discovery of negative components, while interesting, lacks sufficient exploration.
>
> **A:** We'd like to highlight that the primary objective of our work is seeking a principled framework to answer the question of "how do components in diffusion models contribute to a generated concept?", and the "discovery of negative component" is a significant finding, but not a focus, of the paper. Specifically, we propose the first-order, inexpensive approximation to locate important components (specifically, parameters -- the most fine-grained components), confirm the localization hypothesis, and empirically validate the utility of these components; these contributions are novel and important in understanding diffusion models, and distinct from existing interpretability papers.
>
> We agree with the reviewer that the intensive exploration of negative components would be important, but it is beyond the already extensive scope of our work. For example, one can study how to fine-tune these negative components for better editing results, which can be, by itself, a self-contained work. We thus leave these explorations for future work.
>
> **Q4:** Weak Theoretical Foundation.
>
> **A:** In our study, we only focus on a small number of parameters, and ablating them induces an insignificant change in the input $w$ of the objective $J(c,w)$. As discussed in Section 4.2, this assumption makes the first-order approximation *theoretically accurate*, and we empirically validate that in Section 6.1 as well as extensive experiments where we ablate positive and negative components of several concepts on SD-1.4 and SD-2.1.
>
> While theoretical analysis is desirable, both reliance on the well-established first-order approximation (with valid assumptions) and rigorous empirical confirmation of the approximation would strongly justify the soundness of our framework. Coupled with the extensive empirical observation, the framework is expected to generalize to other models and concepts. Nevertheless, we would be happy to address any specific suggestions the reviewer may have regarding the theoretical analysis that the reviewer believes is necessary.
>
> **Q5:** Incomplete Analysis of Negative Components.
>
> **A:** Thank you for your suggestion. As mentioned above, our paper is the *first work* that points out the existence of negative components and their *roles*, i.e., ablating them enhances knowledge in diffusion. We believe that theoretically studying their origin during the training process is an interesting direction, yet it deserves a separate intensive study, and we leave it to future work (please also see our response to Q3 on the paper's objective and contributions).
>
> **Q6:** Narrow Experimental Scope: The evaluation is limited to Stable Diffusion-1.4
>
> **A:** We'd like to clarify that the experiments in our paper follow standard evaluation in the literature and are extensively conducted not only on SD-1.4 (main paper), but also SD-2.1 (appendix). Meanwhile, existing works also only apply their methods on SD-1.4 [2,11,12,13,19,21,25,43], SD-1.5 [5], and SD-2.1[49]. This makes our evaluation already more extensive than these related works.
>
> Nevertheless, we follow the review's suggestion and report the results on SD-3.5 and SD-XL. We ablate no more than 0.03% of the model's parameters and measure the number of nudity content in images generated from prompts in I2P, Ring-A-Bell, and MMA datasets, as well as CLIP-Score on COCO30k. The results show that in different diffusion models, CAD can locate positive components and knowledge is still localized. In particular, CAD reduces the number of nudity labels by 97.39% and 98.63% on SD-XL and jailbreaking prompts such as Ring-A-Bell and MMA datasets, while inducing minimal impact on CLIP-Score. This experiment further indicates the generalization and practicality of our framework.
>
> | Dataset     | Model | Method     | Total         |
> |-------------|-------|------------|---------------|
> | I2P         | SD-3  | Base model | 514           |
> |             |       | CAD        | 157 (-69.46%) |
> |             | SD-XL | Base model | 748           |
> |             |       | CAD        | 95 (-87.30%)  |
> | Ring-A-Bell | SD-3  | Base model | 314           |
> |             |       | CAD        | 30 (-90.45%)  |
> |             | SD-XL | Base model | 537           |
> |             |       | CAD        | 14 (-97.39%)  |
> | MMA         | SD-3  | Base model | 511           |
> |             |       | CAD        | 48 (-90.61%)  |
> |             | SD-XL | Base model | 803           |
> |             |       | CAD        | 11 (-98.63%)  |
>
> | Model  | Method     | CLIP-Score |
> |--------|------------|------------|
> | SD-3.5 | Base model | 32.08      |
> |        | CAD        | 30.63      |
> | SD-XL  | Base model | 31.93      |
> |        | CAD        | 31.67      |
>
> **Q7:** Concept Entanglement Issues.
>
> **A:** The main contribution of CAD is to confirm the localization hypothesis and the existence and utility of negative components, which is significant and hasn't been explored in prior studies. The effect of different concepts and their entanglement, despite being interesting, is an orthogonal direction to our framework and requires an independent study that is beyond the scope of this work.
>
> **Q8:** Lack of Statistical Rigor.
>
> **A:** Our evaluation follows exactly the experimental setup in existing research [5,11,12,13,19,21,25,43,49]. In particular, to evaluate the knowledge of objects and artistic styles, we generate images from several prompts and random seeds *provided in the dataset*. This evaluation helps reduce the randomness in the result. To evaluate the knowledge of nudity content, we generate with a fixed seed on several datasets (I2P, Ring-A-Bell, MMA) with massive prompts (31285 in total) that assess every aspect of the knowledge. This high number of evaluation prompts provides a reliable verification of our claims.
>
> We also evaluate knowledge of nudity content in the base Stable Diffusion, ConceptPrune, and CAD with 5 different random seeds and provide the mean and standard deviation of the number of nudity labels. The table confirms that ablating positive components found by CAD indeed erases the knowledge from the model.
>
> |   Dataset   |     Method    |      Total     |
> |----------|-------------|--------------|
> | I2P         | Baseline      | 699.8 ± 302.9  |
> |             | Concept Prune | 85.4 ± 37.5    |
> |             | CAD           | 55.2 ± 12.8    |
> | Ring-A-Bell | Baseline      | 444.8 ± 44.7   |
> |             | Concept Prune | 99.6 ± 40.1    |
> |             | CAD           | 4.2 ± 1.5      |
> | MMA         | Baseline      | 1715.4 ± 394.5 |
> |             | Concept Prune | 94.8 ± 27.3    |
> |             | CAD           | 113.8 ± 31.2   |

---

> > ### Comment · Reviewer_Yq3z · 2025-08-05
> >
> > I appreciate the additional experiments presented in the rebuttal, as well as the inclusion of error bars in the tables. However, I continue to believe that the contribution of this work remains limited in terms of novelty and the overall scope of the effort.
> >
> > I also support the claim of RK1V about the relevancy of CAM

---

> > > ### Author Response · Authors · 2025-08-05
> > >
> > > We appreciate the reviewer for the response. As explained to Reviewer RK1V about CAM and our motivation/contribution/novelty, CAD is **fundamentally different** in both motivation and methodology to CAM:
> > >
> > > * CAM inspects important features of a **specific input**. While model's params and input features are both nodes in the computational graph, input features are **input dependent**. For example, if certain features are responsible for predicting Picasso's art style in a specific input, ablating those features would alter the model’s prediction for that input; however, this would not necessarily affect the prediction for a different input, as each input may rely on a distinct set of features to support the Picasso prediction.
> > >
> > > * On the other hand, model's parameters are **input independent** -- i.e., model's params are "exactly the same" in different computational graphs of different inputs -- and represent model's knowledge -- i.e., what the model expresses across different inputs. For example, removing the parameters responsible for "Picasso's art style" (as shown in our paper), will induce the model to generate styles other than Picasso, regardless of the input prompts.
> > >
> > > While we appreciate the reviewer's comments regarding "the limited novelty and scope" of our work, we are also unsure about the reasons leading to these conclusions. As explained in the rebuttal (with references to corresponding sections in the paper), we focus on parameters (instead of layers as in existing work), propose an efficient estimator for diffusion model, reveal new insights of how diffusion models generate concepts (existence of positive and negative components), and extensively validate CAD on multiple datasets and models following existing evaluation protocal in the original paper while providing new results in the rebuttal based on the reviewer's suggestions. We'd be grateful if the reviewer could elaborate on which of these contributions are perceived as limited, so that we can improve our work.

---

### Official Review · Reviewer_UcXL · 2025-06-28

**Clarity:** 3
**Significance:** 2
**Originality:** 3
**Rating:** 5
**Confidence:** 3

**Summary:**

This paper introduces Component Attribution for Diffusion Model (CAD), an efficient framework for interpreting how parameters in diffusion models generate concepts. The study reveals that concepts are governed by localized "positive components" that induce them, and uniquely, "negative components" that suppress them. Based on these findings, the paper presents two lightweight, inference-time model editing algorithms: CAD-Erase, which removes concepts by ablating positive components, and CAD-Amplify, which enhances concepts by ablating negative ones. Extensive experiments validate that this approach can effectively and robustly erase or amplify objects, styles, and explicit content while preserving other knowledge.

**Questions:**

1.On Cost: Please quantify the practical overhead (inference speed, memory) compared to baseline.

2.On Generalizability: How does the "localization hypothesis" generalize to newer architectures like SD3 or FLUX? We suggest adding theoretical reasoning or a preliminary analysis.

3.On Optimization: Could the method of directly setting ablating parameters to zero be optimized other than finetuning?

**Ethical Concerns:**

["NO or VERY MINOR ethics concerns only"]

**Final Justification:**

Most of my concerns have been resolved. I will raise the rating.

**Limitations:**

yes, authors addressed the limitations and potential negative societal impact of their work

**Quality:**

2

**Strengths And Weaknesses:**

Strengths:
- State-of-the-Art Performance: Achieves SOTA or highly competitive results across diverse benchmarks.
- Demonstrated Robustness to Attacks: The paper shows that CAD is more robust to white-box attacks than many competing methods.

Weaknesses:
- No Overhead Analysis: Lacks analysis of the practical computational cost (time, memory) for editing and its effect on the final model's inference speed.
- Uncertain Generalizability: Experiments are confined to Stable Diffusion 1.4(mostly) and 2.1, leaving its effectiveness on other architectures (like SDXL, SD3) unproven.
- Crude Intervention Method: The approach of ablating parameters by setting them to zero is a blunt instrument. While effective, it might not be optimal.

---

> ### Author Rebuttal · Authors · 2025-07-31
>
> We are grateful for your helpful comments. Please see our response below.
>
> **Q1:** No Overhead Analysis
>
> **A:** We'd like to clarify that CAD as well as other erasing methods edit diffusion models by only updating their weights, and this process is performed a single time only. Therefore, CAD  has no impact on inference speed and memory. Furthermore, CAD does not require training, unlike other methods such as ESD; thus, the time and memory overhead for editing is insignificant.
>
> Nevertheless, we follow the review's suggestion and measure the computational cost for editing the model. We conduct experiments on a single A5000 GPU. We observe that ESD takes a significant amount of time since it requires training for 1000 epochs.  ConceptPrune requires generating several images to collect activations for a single concept, leading to high editing time. UCE and RECE offer fast editing due to the closed-form update; however, for a certain type of knowledge such as artistic styles, they require additional computation to preserve many unrelated knowledge. Meanwhile, CAD consumes a small amount of computation, enabling scalability to larger models while maintaining **zero** impact on final inference speed.
>
> | Method            | Object                     |                    | Nudity                     |                    | Artist                     |                    |
> |-------------------|----------------------------|--------------------|----------------------------|--------------------|----------------------------|--------------------|
> |                   | Time (sec) | Memory (MB) | Time (sec) | Memory (MB) | Time (sec) | Memory (MB) |
> | **ESD**           | 5,691       | 10,225       | 5,624       | 10,233       | 5,617       | 10238       |
> | **ConceptPrune** | 230        | 4,047        | 294        | 4,075        | 321        | 4035        |
> | **RECE**          | 26         | 15,561       | 66         | 15,562       | 470        | 15665       |
> | **UCE**           | 9          | 7,420        | 19         | 7,658        | 386        | 8610        |
> | **CAD**           | 66         | 8,152        | 65         | 8,157        | 66         | 8223        |       |      |
>
>
> **Q2:** Experiments are confined to Stable Diffusion 1.4(mostly) and 2.1, leaving its effectiveness on other architectures (like SDXL, SD3) unproven. How does the "localization hypothesis" generalize to newer architectures like SD3 or FLUX? We suggest adding theoretical reasoning or a preliminary analysis.
>
> **A:** We'd like to note that our experimental setup follows existing works that apply their methods on SD-1.4 [2,11,12,13,19,21,25,43], SD-1.5 [5], and SD-2.1 [49], making our evaluation quite extensive compared to these works. Nevertheless, we follow the review's suggestion and report the results on SD-3.5 and SD-XL. We ablate no more than 0.03% of the model's parameters and measure the number of nudity content in images generated from prompts in I2P, Ring-A-Bell, and MMA datasets, as well as CLIP-Score on COCO30k. The results show that in different diffusion models, CAD can locate positive components, and knowledge is still localized. In particular, CAD reduces the number of nudity labels by 97.39% and 98.63% on SD-XL and jailbreaking prompts such as Ring-A-Bell and MMA datasets, while inducing minimal impact of CLIP-Score. This experiment further indicates the generalization and practicality of our framework.
>
> Regarding the theoretical reasoning of the localization hypothesis, we believe that it could be a result of the training process on over-parameterized models. Theoretically studying its origin is an interesting direction, yet it deserves a separate intensive study, and we leave it to future work.
>
>
> | Dataset     | Model | Method     | Armpits | Belly | Buttocks | Feet | Breast (F) | Genitalia (F) | Breast (M) | Genitalia (M) | Anus | Total |
> |-------------|-------|------------|---------|-------|----------|------|------------|---------------|------------|---------------|------|-------|
> | I2P         | SD-3  | Base model |     158 |   130 |       19 |   41 |        137 |             3 |         16 |            10 |    0 |   514 |
> |             |       | CAD        |      30 |    32 |        3 |    2 |         64 |             5 |         10 |            11 |    0 |   157 (-69.46%) |
> |             | SD-XL | Base model |     222 |   169 |       32 |   44 |        211 |            14 |         42 |            14 |    0 |   748 |
> |             |       | CAD        |      46 |     9 |        2 |    7 |         19 |             2 |          1 |             9 |    0 |    95 (-87.30%) |
> | Ring-A-Bell | SD-3  | Base model |      66 |    73 |        6 |   30 |        110 |             4 |         25 |             0 |    0 |   314 |
> |             |       | CAD        |       6 |     7 |        0 |    0 |         14 |             0 |          3 |             0 |    0 |    30 (-90.45%) |
> |             | SD-XL | Base model |     106 |   110 |       19 |   33 |        203 |            50 |         16 |             0 |    0 |   537 |
> |             |       | CAD        |       3 |     1 |        0 |    0 |         10 |             0 |          0 |             0 |    0 |    14 (-97.39%) |
> | MMA         | SD-3  | Base model |     216 |    96 |       51 |   21 |         68 |             1 |         49 |             9 |    0 |   511 |
> |             |       | CAD        |       8 |    13 |        6 |    3 |          7 |             2 |          5 |             4 |    0 |    48 (-90.61%) |
> |             | SD-XL | Base model |     209 |   185 |       47 |   75 |        212 |            11 |         52 |            12 |    0 |   803 |
> |             |       | CAD        |       0 |     2 |        0 |    6 |          0 |             0 |          0 |             3 |    0 |    11 (-98.63%) |
>
> | Model  | Method     | CLIP-Score |
> |--------|------------|------------|
> | SD-3.5 | Base model | 32.08      |
> |        | CAD        | 30.63      |
> | SD-XL  | Base model | 31.93      |
> |        | CAD        | 31.67      |
>
>
> **Q3:** The approach of ablating parameters by setting them to zero is a blunt instrument. While effective, it might not be optimal. Could the method of directly setting ablating parameters to zero be optimized other than finetuning?
>
> **A:** Thank you for the interesting suggestion.
> Our work aims to answer the research question "how do components in diffusion models contribute to a generated concept?" -- thus, we focus on directly ablating the identified components as a means to validate their roles. More specifically, CAD can locate positive and negative components of a knowledge, confirmed by their ability to "edit" knowledge in diffusion models via ablation. Importantly, the implications of our work go beyond ablation; for example, as you suggested, existing finetuning unlearning approaches could potentially be combined with these components to further improve the unlearning ability. We leave these promising applications as future extensions of our work.

---

> > ### Comment · Reviewer_UcXL · 2025-08-06
> >
> > Thanks for the reply. Most of my concerns have been resolved. I will consider raise the rating in the final decision.

---

> > > ### Author Response · Authors · 2025-08-07
> > >
> > > We really appreciate your constructive feedback and hope that the reviewer will maintain a supportive reception of our work. We will include these valuable discussions in the camera-ready version of the paper. If you have any other concerns or suggestions, we are happy to discuss further.

---

### Official Review · Reviewer_RK1V · 2025-06-30

**Clarity:** 3
**Significance:** 2
**Originality:** 2
**Rating:** 4
**Confidence:** 5

**Summary:**

This paper proposes **Component Attribution for Diffusion Models (CAD)**, a method for understanding and analyzing generative diffusion models. The core approach identifies parameters within the model that have a positive or negative influence on specific concepts during generation. The authors posit that only a small subset of parameters are relevant to a given concept. Based on this attribution analysis, they introduce two corresponding algorithms designed to **suppress or enhance** the generation of the target concept. The fundamental attribution mechanism involves **perturbing a parameter \(w\) (e.g., setting it towards zero)** and measuring the resulting impact on the presence of concept \(c\) in the output image \(x\). The effect on concept \(c\) is quantified using a **discriminative model**.

**Questions:**

None

**Ethical Concerns:**

["NO or VERY MINOR ethics concerns only"]

**Final Justification:**

Most of my concerns have been resolved. However, I still believe the novelty is somewhat limited. Indeed, CAM modifies input features, this paper modifies 'components'. However, both input and parameters are simply nodes on the computational graph. And there are many works have also investigated the change of parameters. On the other hand, I admit that the authors did provide many efforts on empirical evaluations. Maybe there are researchers that can benefit from the results of this work. Considering all the facts, I decide to raise my final score to 4 but not against for rejection.

**Limitations:**

The authors did not provide such analyses.

**Quality:**

3

**Strengths And Weaknesses:**

### **Strengths**
1.  **Engaging Topic:** The analysis of internal mechanisms in Diffusion Models is a highly relevant and important research direction.
2.  **Solid Background:** The related works section is well-surveyed, establishing a solid context for the addressed problem.
3.  **Logical Progression:** The method development follows a logically sound, theoretically grounded progression from attribution analysis to practical application (suppression/enhancement).

### **Weaknesses**
1.  **Unclear Illustrative Figure:** Figure 1 does not effectively illustrate the *method* itself; it primarily presents a *conclusion* (attribution scores), offering limited help in understanding the CAD procedure.
2.  **Insufficient and Underwhelming Experiments:**
    *   Experimental validation is notably limited.
    *   Results lack robustness: Table 1 fails to highlight superior metrics effectively (e.g., via bolding).
    *   Evaluation is primarily conducted on **low-resolution images**.
    *   Model Scope Limitation: Only Stable Diffusion 1.4 (SD v1.4) is tested. The conclusions may not generalize to **more advanced and widely used versions like SD2.x or SD3**, which is a significant omission.
    *   Poor Empirical Performance: Qualitative results are weak. For instance, Figure 3 (rows 1-2) shows that concept "erase" significantly alters background information, indicating the method lacks practical advantage and suffers from unintended side effects.
3.  **Limited Methodological Novelty:** The core attribution technique (perturbing parameters and measuring output change via a classifier) is **fundamentally simple** and bears strong resemblance to well-established gradient/importance-based attribution methods developed over the past decade (e.g., CAM and its numerous derivatives). The novelty contribution is therefore **significantly lacking**.
4.  **Misleading Terminology:**
    *   Referring to individual *parameters* as "Components" is inaccurate and potentially confusing. "Component" typically implies a composed module or functional unit within a network, not a single scalar weight.
    *   Labeling a *class token* or a *prompt term* as a "concept" is inappropriate. In the field, "concept" generally denotes a higher-level, often abstract, semantic entity (e.g., learned by adapters like LoRA), distinct from the direct input tokens used here.

---

> ### Author Rebuttal · Authors · 2025-07-31
>
> Thank you for your insightful feedback. Please see our response below.
>
> **Q1:** Figure 1 does not effectively illustrate the method itself; it primarily presents a conclusion (attribution scores), offering limited help in understanding the CAD procedure.
>
> **A:** Fig. 1 illustrates the "problem statement" of our work: there exist negative and positive components for each concept, where their ablations result in "increasing" or "decreasing", respectively, the presence of that concept in the generated image. Thus, it does not illustrate the specific technique used to **identify these components**; please note that the specific proposed algorithms are detailed in Algorithms 1 and 2. We believe this division is clearer to highlight the studied research problem and the proposed algorithmic solution.
>
> **Q2:** Experimental validation is notably limited.
>
> Thank you for the comment. However, we'd appreciate it if the reviewer could elaborate the "limited experimental validation", as it'd better help us to address your concerns.
>
> In the paper, we conduct experiments on SD-1.4 and SD-2.1, 10 objects, 5 artist styles, nudity content, 2 black-box attacks, and 2 white-box attacks.  Our experimental validation is extensive, following the setting in prior research [5,11,12,13,19,21,25,43].
>
> **Q3:** Results lack robustness: Table 1 fails to highlight superior metrics effectively (e.g., via bolding).
>
> **A:** We will highlight the best performance method in Table 1 in the final version. We'd also like to note that the main goal of our work is to study the localization hypothesis in diffusion models, rather than proposing a state-of-the-art knowledge erasing method. Table 1 along with other empirical evidence in the paper successfully verify that hypothesis by showing that ablating a small number of parameters removes the target concept while keeping other concepts intact. We believe that future research could utilize our findings to improve knowledge erasing algorithms.
>
>
> **Q4:** Evaluation is primarily conducted on low-resolution images.
>
> **A:** We'd like to clarify that we generate images using **the standard resolution of Stable Diffusion models**, including 512x512 and 768x768 for SD-1.4 and SD-2.1, respectively. These are high resolutions and have also been used in several related studies and benchmarks [5,11,12,13,19,21,25,43,49]. Moreover, we also conduct experiments with image size of 1024x1024 on SD-3.5 and SD-XL below, further confirming the utility of our framework.
>
> **Q5:** Model Scope Limitation: Only Stable Diffusion 1.4 (SD v1.4) is tested
>
> **A:** We'd like to clarify that the experiments in our paper are extensively conducted not only on SD-1.4 (main paper), but also SD-2.1 (appendix), which is mentioned in the reviewer's comment. Meanwhile, existing works also only apply their methods on SD-1.4 [2,11,12,13,19,21,25,43], SD-1.5 [5], and SD-2.1[49].
>
> We believe these experiments are sufficiently comprehensive to support our claim and demonstrate the utility of our framework.
>
> Nevertheless, we follow the review's suggestion and report the results on SD-3.5 and SD-XL. We ablate no more than 0.03% of the model's parameters and measure the number of nudity content in images generated from prompts in I2P, Ring-A-Bell, and MMA datasets, as well as CLIP-Score on COCO30k. The results show that in different diffusion models, CAD can locate positive components, and knowledge is still localized. In particular, CAD reduces the number of nudity labels by 97.39% and 98.63% on SD-XL and jailbreaking prompts such as Ring-A-Bell and MMA datasets, while inducing minimal impact on CLIP-Score. This experiment further indicates the generalization and practicality of our framework.
>
> | Dataset     | Model | Method     | Armpits | Belly | Buttocks | Feet | Breast (F) | Genitalia (F) | Breast (M) | Genitalia (M) | Anus | Total |
> |-------------|-------|------------|---------|-------|----------|------|------------|---------------|------------|---------------|------|-------|
> | I2P         | SD-3  | Base model |     158 |   130 |       19 |   41 |        137 |             3 |         16 |            10 |    0 |   514 |
> |             |       | CAD        |      30 |    32 |        3 |    2 |         64 |             5 |         10 |            11 |    0 |   157 (-69.46%) |
> |             | SD-XL | Base model |     222 |   169 |       32 |   44 |        211 |            14 |         42 |            14 |    0 |   748 |
> |             |       | CAD        |      46 |     9 |        2 |    7 |         19 |             2 |          1 |             9 |    0 |    95 (-87.30%) |
> | Ring-A-Bell | SD-3  | Base model |      66 |    73 |        6 |   30 |        110 |             4 |         25 |             0 |    0 |   314 |
> |             |       | CAD        |       6 |     7 |        0 |    0 |         14 |             0 |          3 |             0 |    0 |    30 (-90.45%) |
> |             | SD-XL | Base model |     106 |   110 |       19 |   33 |        203 |            50 |         16 |             0 |    0 |   537 |
> |             |       | CAD        |       3 |     1 |        0 |    0 |         10 |             0 |          0 |             0 |    0 |    14 (-97.39%) |
> | MMA         | SD-3  | Base model |     216 |    96 |       51 |   21 |         68 |             1 |         49 |             9 |    0 |   511 |
> |             |       | CAD        |       8 |    13 |        6 |    3 |          7 |             2 |          5 |             4 |    0 |    48 (-90.61%) |
> |             | SD-XL | Base model |     209 |   185 |       47 |   75 |        212 |            11 |         52 |            12 |    0 |   803 |
> |             |       | CAD        |       0 |     2 |        0 |    6 |          0 |             0 |          0 |             3 |    0 |    11 (-98.63%) |
>
> | Model  | Method     | CLIP-Score |
> |--------|------------|------------|
> | SD-3.5 | Base model | 32.08      |
> |        | CAD        | 30.63      |
> | SD-XL  | Base model | 31.93      |
> |        | CAD        | 31.67      |
>
> **Q6:**  Qualitative results are weak. For instance, Figure 3 (rows 1-2) shows that concept "erase" significantly alters background information, indicating the method lacks practical advantage and suffers from unintended side effects.
>
> **A:** We'd first like to clarify that the qualitative results are intended to demonstrate the successful removal of the concepts, while showing minimal interference with other concepts. Extensive qualitative results on SD-1.4 (main paper) and SD-2.1 (appendix) verify this property.
>
> Second, as the diffusion process involves multiple sampling steps, slight variations in the generated images (or their backgrounds) are expected in multiple runs (those with or without ablations). This behavior is consistent across all methods, where the backgrounds change in multiple generations. Nevertheless, the key focus is the successful removal (or reservation) of the concept.
>
> Finally, the quantitative results, based on well-established evaluation processes, clearly demonstrate the performance of the baselines and CAD. In other words, the combination of qualitative and quantitative results consistently highlights the effectiveness of CAD.
>
>
> **Q7:** Limited Methodological Novelty: The core attribution technique (perturbing parameters and measuring output change via a classifier) is fundamentally simple and bears strong resemblance to well-established gradient/importance-based attribution methods developed over the past decade (e.g., CAM and its numerous derivatives). The novelty contribution is therefore significantly lacking.
>
> **A:** As discussed in the Related works, our paper attributes *model components*, which is fundamentally different from prior works such as CAM that attribute the *input features*. Section 4.2 also highlights the significant challenge in model (component) attribution for diffusion (Section 4.2); our proposed algorithm offers a significantly efficient solution to this problem, making it a valuable contribution to the literature.
>
> More importantly, the primary contribution of this work extends beyond the efficient solution itself -- it provides significant insights into the role of model components in diffusion models. Through CAD, we confirm the localization hypothesis (Section 6.2) and reveal the existence of negative components (Section 6.3), which are missing in prior works.
>
> **Q8:** Misleading Terminology
>
> **A:** In this work, we refer to "components" as grouped model parameters [34], such as convolutional filters, attention heads, layers, etc. This definition allows us to work with different levels of granularity, from coarse-grained components such as layers in existing works [2,3] or fine-grained components such as parameters in our paper.
> Regarding the use of "concept", we follow the widely accepted definition that has been used in prior works [5,11,12,13,19,21,25,43,49].

---

> ### Comment · Reviewer_RK1V · 2025-08-04
>
> Thanks for the detailed reply. Most of my concerns have been resolved. However, I still believe the novelty is somewhat limited. Indeed, CAM modifies input features, this paper modifies 'components'. However, both input and parameters are simply nodes on the computational graph. And there are many works have also investigated the change of  parameters. On the other hand, I admit that the authors did put effort on empirical evaluations. Maybe there are researchers that can benefit from the results of this work. Considering all the facts, I decide to raise my final score to 4 but not against for rejection.

---

> > ### Author Response · Authors · 2025-08-05
> >
> > We are grateful that our paper receives your positive support. Regarding the novelty of our work, we'd like to clarify again that our framework and CAM are fundamentally different in both **motivation**, and **methodology**:
> >
> > * CAM inspects important features of a **specific input**. While model's params and input features are both nodes in the computational graph, input features are **input dependent**. For example, if certain features are responsible for predicting Picasso's art style in a specific input, ablating those features would alter the model’s prediction for that input; however, this would not necessarily affect the prediction for a different input, as each input may rely on a distinct set of features to support the Picasso prediction.
> >
> > * On the other hand, model's parameters are **input independent** -- i.e., model's params are "exactly the same" in different computational graphs of different inputs -- and represent model's knowledge -- i.e., what the model expresses across different inputs. For example, removing the parameters responsible for "Picasso's art style" (as shown in our paper), will induce the model to generate styles other than Picasso, regardless of the input prompts.
> >
> > * Although there are existing works studying model's parameters, these approaches are significantly challenging to apply in diffusion models as discussed in detail in Section 4.2. In contrast, CAD offers an efficient and effective way to estimate the contribution of parameters.
> >
> > We hope that this clarification and the accompanying examples more clearly illustrate the motivation and contributions of CAD, and we hope the reviewer will maintain a supportive reception of our work.

---

### Official Review · Reviewer_tDav · 2025-07-03

**Clarity:** 3
**Significance:** 3
**Originality:** 2
**Rating:** 5
**Confidence:** 4

**Summary:**

This paper describes the use of a linear counterfactual predictor to attibute concept-related capabilities to sets of diffusion model components. It identifies components with activations magnitudes have both positive and negative relationships with the generation of images containing the concept. The component outputs are then modulated to amplify or attenuate the concept.

**Questions:**

If the contributions were revised to focus on the specific novel contributions of the paper, what would they look like?

Do you claim to be the first to discover that a very small percent of DNN components have nearly all the influence over the expression of a concept?

Is the key novel idea for the counterfactual predictor to use the Taylor series expansion to enable efficient prediction, or is there another primary novel contribution related to the predictor?

Do you claim to be the first to find that there are some components for which higher activations repress a concept?

Are there other novel ideas or discoveries claimed?

**Ethical Concerns:**

["NO or VERY MINOR ethics concerns only"]

**Final Justification:**

The rebuttals suggest that the final version of the paper will be clearer w.r.t. novelty claims.

**Limitations:**

Yes.

**Paper Formatting Concerns:**

Adequate.

**Quality:**

3

**Strengths And Weaknesses:**

Strengths

The approach is highly effective based the experimental evaluation, which is adequate in my opinion.

The results provide additional evidence that "knowledge is localized", although this observation has been made in prior work on attribution and on fairness.

The authors claim that this is the first work commenting on the existence of and identifying components whose activations are inversely related to the expression of concepts. A brief literature survey supports this claim. This is an important finding that others in the field should  be aware of.

The presented attribution method requries only a single forward and backward pass in contrast with approaches generating data to learn $\alpha_c$.

Weaknesses

The writing quality is a bit poorer than typical for a NeurIPS submission.

Design decisions are not very well justified, relative to alternatives. For example, the authors make clear that the first-order Taylor series expansion enables a efficient estimation but do not comment on why their chosen approach is likely to do a better job of this than alternatives. More explanation of the reasons for design decisions would improve the paper.

---

> ### Author Rebuttal · Authors · 2025-07-31
>
> We are grateful for your insightful comments and for appreciating the effectiveness of our method and the importance of our findings. Please see our response below.
>
> **Q1:** The writing quality is a bit poorer than typical for a NeurIPS submission.
>
> **A:** Thank you for your comment. We will update the writing in the camera-ready version according to the discussion during the rebuttal.
>
> **Q2:** Design decisions are not very well justified, relative to alternatives. For example, the authors make clear that the first-order Taylor series expansion enables an efficient estimation but do not comment on why their chosen approach is likely to do a better job of this than alternatives. More explanation of the reasons for design decisions would improve the paper.
>
> **A:** Thank you for your comment. The alternatives are actually impractical for diffusion models, as discussed in Section 4.2, due to the data construction process. In particular, creating one training sample for the linear estimator requires computing the objective function on the masked model, and we need a high number of samples to train an accurate estimator. For example, [34] constructs 100,000 samples for ResNet50 with 25M parameters; with 859M parameters of SD-1.4, the computational overhead of training the linear estimator becomes extremely high, if not impractical.
>
> Furthermore, both CAD and the linear estimator approach are an approximation of the true objective function, based on the linearity assumption. Section 6.1 shows that CAD yields a high correlation and can approximate the true objective with high accuracy.
>
> For these reasons, CAD is a much more desirable choice than the existing alternatives.
>
> **Q3:** If the contributions were revised to focus on the specific novel contributions of the paper, what would they look like?
>
> Do you claim to be the first to discover that a very small percent of DNN components have nearly all the influence over the expression of a concept?
>
> Is the key novel idea for the counterfactual predictor to use the Taylor series expansion to enable efficient prediction, or is there another primary novel contribution related to the predictor?
>
> Do you claim to be the first to find that there are some components for which higher activations repress a concept?
>
> Are there other novel ideas or discoveries claimed?
>
> **A:** We'd like to clarify our contributions as follows. First, we propose a framework that helps identify the contribution of model components to generated data. Although the idea of using counterfactual estimator and Taylor approximation has been explored in classification and language models [34,38], applying them to diffusion is challenging due to the reverse process. Our work shows that we can design a suitable objective function such that the approximation is accurate and becomes beneficial for the study in next sections.
>
> Secondly, we utilize our framework to confirm the localization hypothesis. Prior study focuses on coarse-grained components such as layers and conclude that knowledge is distributed amongst UNet and localized in the text encoder. In contrast, we study fine-grained components and pinpoint a small number of parameters that highly contribute to a knowledge, indicating that knowledge is localized.
>
> Finally, we are the first that observe and verify the existence of negative components in diffusion models. These components contribute negatively to a knowledge; in particular, ablating them leads to high generation probability.

---

> > ### Comment · Reviewer_tDav · 2025-08-04
> > **Thank you for your rebuttals**
> >
> > The rebuttals are responsive to my questions and give me some confidence that the clarity of the writing w.r.t. contributions will be improved in the final version of the paper.

---

> > > ### Author Response · Authors · 2025-08-05
> > >
> > > We'd like to thank the reviewer for positively supporting our paper.

---

### Comment · Area_Chair_1db6 · 2025-08-04
**Author-reviewer discussion period ends soon**

Dear reviewers,

Author-reviewer discussion period ends soon. Please check the rebuttals and take an appropriate action.

AC

---

### Author Response · Authors · 2025-08-09

We greatly appreciate the reviewers for their insightful feedback. During the rebuttal, we have:

- Clarified the contribution and novelty of our work in the response to Reviewer tDav, RK1V, and Yq3z. Specifically, our work (1) studies the most fine-grained parts of the model (i.e., parameters), (2) proposes an efficient model attribution framework that helps verify the localization hypothesis, and (3) identifies the existence of negative components in diffusion models. Our framework also offers two quick model editing methods. The importance of our findings is acknowledged by Reviewer tDav.
- Clarified that our framework and CAM are fundamentally different in both motivation and methodology in the response to Reviewer RK1V and Yq3z. Specifically, CAM is input-specific while our framework is input-independent, allowing us to locate the internal knowledge in diffusion. Unfortunately, due to the time limit, the reviewers couldn't engage with us further.
- Provided additional experiments on SD-3.5 and SD-XL in the response to Reviewer RK1V, UcXL and Yq3z. These results, along with extensive quantitative and qualitative evaluation in the original version (which already closely followed existing protocols), further demonstrate the effectiveness of our framework, as acknowledged by Reviewer tDav and UcXL.
- Clarified the purpose of Figure 1 and the use of "component" and "concept" terminology in the response to Reviewer RK1V.
- Provided the analysis on computational cost and time in the response to Reviewer UcXL.
- Discussed other potential editing strategy in the response to Reviewer UcXL.
- Clarified the theoretical foundation of our approach in the response to Reviewer Yq3z.
- Provided additional statistical results in the response to Reviewer Yq3z.

We hope that our responses have addressed the concerns of the reviewers. Finally, we're happy to promptly answer additional questions.

---

### Note · Authors · 2025-08-13

We would like to express our gratitude to the reviewers for their time and engagement in the discussion, and for providing thoughtful feedback and constructive comments. During the discussion time, we have carefully addressed all of the concerns raised regarding the novelty and clarity, and provided additional experiments on other models. Please see the general official comment for more details. We hope that you could consider this in the final recommendation.

Once again, thank you for your dedication and contribution to the community.

---

### Decision · Program_Chairs · 2025-09-17

**Decision:**

Accept (poster)

**Comment:**

The paper proposes Component Attribution for Diffusion Models (CAD), a framework for interpreting and editing diffusion models at the parameter level. CAD identifies positive components (promoting concept generation) and negative components (suppressing concepts). Leveraging these, the authors introduce two inference-time editing algorithms: CAD-Erase (to remove concepts) and CAD-Amplify (to enhance them).

The topic is timely, and better understanding inner workings of diffusion model is important. Reviewer scores and opinions are spread. As for the weaknesses, the method itself is not very novel and the experiment/analysis of the CAD-Erase and CAD-Amplify can be improved as clearly pointed out by reviewer Yq3z. Nonetheless, the findings on negative components and observation of localization can be useful for the research community and the other three reviewers concluded with positive scores after extensive rebuttals. Therefore, an acceptance is recommended.